# PnP-Corrector: A Universal Correction Framework for Coupled Spatiotemporal Forecasting

**Hao Wu** [* 1 2]  **Fan Xu** [* 3]  **Yuxu Lu** [4]  **Penghao Zhao** [5]  **Fan Zhang** [6]  **Hao Jia** [2 5]  **Yuxuan Liang** [7]  **Ruijian Gou** [8]
**Qingsong Wen** [9]  **Xian Wu** [2]  **Xiaomeng Huang** [1]  **Yuan Gao** [1]

## Abstract

Coupled spatiotemporal forecasting is important for predicting the future evolution of multiple interacting dynamical systems, such as in climate models. However, existing methods are severely constrained by the persistent bottleneck of compounding errors. In coupled systems, errors from each subsystem simulator propagate and amplify one another, a phenomenon we term *Reciprocal Error Amplification* leading to a rapid collapse of long-range predictions. To address this challenge, we propose a universal framework called `PnP-Corrector` (**P**lug-a**n**d-**P**lay **Corrector**). The core idea of our framework is to decouple the physical simulation from the error correction process: it freezes pre-trained physics simulation engines and exclusively trains a correction agent to proactively counteract the systematic biases emerging from the coupled system. Furthermore, we design an efficient predictive model architecture, `DSLCast`, to serve as the backbone of this framework. Extensive experiments demonstrate that our method significantly enhances the long-term stability and accuracy of coupled forecasting systems. For instance, in the challenging task of a 300-day global ocean-atmosphere coupled forecast, our `PnP-Corrector` framework reduces the prediction error of the baseline model by 28% and surpasses state-of-the-art models on several key metrics. Codes link: `https://github.com/Alexander-wu/PnP-Corrector`.

---

[*]Equal contribution [1]Tsinghua University [2]Tencent [3]Shenzhen Loop Area Institute [4]The Hong Kong Polytechnic University [5]Peking University [6]The Chinese University of Hong Kong [7]The Hong Kong University of Science and Technology (Guangzhou) [8]Ocean University of China [9]Squirrel Ai Learning. Correspondence to: Xian Wu <kevinxwu@tencent.com>, Xiaomeng Huang <hxm@tsinghua.edu.cn>, Yuan Gao <yuangao24@mails.tsinghua.edu.cn>.

*Proceedings of the 43rd International Conference on Machine Learning*, Seoul, South Korea. PMLR 306, 2026. Copyright 2026 by the author(s).

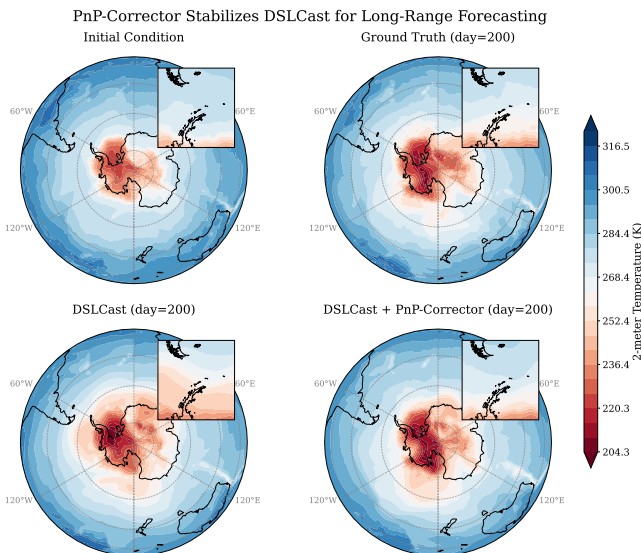

PnP-Corrector Stabilizes DSLCast for Long-Range Forecasting

*Figure 1.* **The `PnP-Corrector` framework enables long-term stability in coupled forecasting.** This figure compares a 200-day 2-meter temperature (T2M) forecast initialized from the state shown top-left. While the standard `DSLCast` baseline (bottom-left) accumulates significant errors and drifts from the true state, our `PnP-Corrector` framework (bottom-right) effectively corrects these systematic biases, producing a forecast that remains in close agreement with the ground truth (top-right).

## 1. Introduction

Spatiotemporal forecasting is a fundamental challenge in computer vision (Gao et al., 2022c; Wu et al., 2024; Shi et al., 2015; Han et al., 2022; Mohajerin & Waslander, 2019) and scientific computing (Zhang et al., 2023; Xiong et al., 2023; Gao et al., 2025b; Wu et al., 2025c). Its goal is to predict the future evolution of dynamical systems, such as video sequences or physical phenomena. Autoregressive models are a dominant paradigm for this task. However, their performance is limited by a persistent bottleneck: ***compounding errors***. Small single-step errors in the prediction process are fed back as input for the next step (Kirkpatrick, 2014; Wenninghoff & Schwammberger, 2025), causing errors to accumulate and leading to significant divergence from the ground-truth trajectory over long horizons.

This challenge is dramatically amplified in Coupled Dy-

namical Systems. In such systems, two or more independent autoregressive simulators, for example, an atmosphere model (Bi et al., 2023; Gao et al., 2025b) and an ocean model (Xiong et al., 2023; Cui et al., 2025) must drive each other in a closed loop. This bidirectional interaction creates a vicious cycle, we term ***Reciprocal Error Amplification****: the output error from one simulator becomes input noise for another, which in turn magnifies the error in the first simulator* (see Figure 2). This exponential crosstalk of errors causes existing coupled systems to collapse after a few iterations, making stable and reliable long-term forecasting an unsolved problem (Lippe et al., 2023; Wu et al., 2025c).

Current strategies to address this challenge have significant limitations. End-to-end fine-tuning of the entire system often disrupts the valuable physical priors learned by each component during pre-training, leading to catastrophic forgetting (Cai et al., 2021; Wang & Yu, 2021; Hess et al., 2022). Crucially, such fine-tuning tends to overfit the model to correcting errors only within a specific, short pre-defined rollout, causing the system to collapse again once the forecast horizon extends beyond this fine-tuned range (Chen et al., 2023; Gao et al., 2025b; Bi et al., 2023; Zhang et al., 2023; Kochkov et al., 2024). On the other hand, simple statistical bias correction methods are inherently static and cannot capture the dynamic error patterns that emerge from the complex interactions within the coupled system (Maraun, 2016; Gawronski, 2004).

To solve this problem, we propose a dual contribution. First, at the architectural level, we design a new, simple, yet powerful predictive model called `DSLCast`. It is based on efficient axially gated convolutions and a differentiable semi Lagrangian advection, making it an ideal building block for complex physical simulators. Second, at the framework level, we propose a model-agnostic correction framework, `PnP-Corrector`, based on the core idea of decoupling simulation from error correction. `PnP-Corrector` is a universal framework for coupled systems and can be expanded to more spheres. For instance, in ocean-atmosphere coupling task, we demonstrate the remarkable flexibility of the `DSLCast` architecture: we use two independent `DSLCast` instances as *physics engines* to predict the ocean and atmosphere, respectively. Innovatively, we use a third `DSLCast` instance as a *Correction Agent*. During coupled forecasting, we freeze the *physics engines* and train only the *Correction Agent*, which learns to counteract the systematic biases of the entire system.

We validate our approach on the highly challenging task of global ocean-atmosphere coupled forecasting and a case study on ocean-atmosphere-land coupled forecasting task. The system we build, using `DSLCast` for all components, achieves unprecedented long-term stability, as visually demonstrated in Figure 1, where our framework corrects the

baseline's significant predictive drift and maintains high fidelity to the ground truth over a 200-day forecast. Crucially, to demonstrate the generality of our framework, we further show that it seamlessly integrates with other state-of-the-art foundation models (e.g., GraphCast (Lam et al., 2023), SimVP (Tan et al., 2025)) and provides them with the same long-term stability. Our contributions are as follows:

➠ ***New Problem Formulation.*** We are the first to identify and formalize the problem of *Reciprocal Error Amplification* in coupled spatiotemporal forecasting.

➠ ***Superior Performance.*** We design `DSLCast`, an efficient and flexible base predictive architecture, and it achieves state-of-the-art performance in coupled spatiotemporal forecasting for Earth systems.

➠ ***Novel Framework.*** We propose `PnP-Corrector`, a model-agnostic, **plug-and-play Correction Agent** framework that provides a new paradigm for stabilizing coupled systems by decoupling simulation and correction.

## 2. Related Work

### 2.1. Spatiotemporal Forecasting Models

Spatiotemporal forecasting is a core problem in computer vision, with applications such as future video frame prediction (Wang et al., 2022; Shi et al., 2015) and human motion forecasting (Wang et al., 2021; Mao et al., 2022; Tanke et al., 2023). Early works (Wu et al., 2024; 2025d; 2026) often rely on Recurrent Neural Network (RNN) architectures like ConvLSTM (Shi et al., 2015), which use recurrent units to capture time dependencies in sequential data. As deep learning evolves, architectures based on Convolutional Neural Networks (CNNs) and Transformers become the mainstream (Gao et al., 2022c; Ronneberger et al., 2015). 3D CNNs (Gao et al., 2022b; Chen et al., 2021) can directly process spatiotemporal data cubes, while the Transformer (Vaswani et al., 2017; Dosovitskiy, 2020) architecture shows excellent ability in capturing long-range dependencies with its powerful self-attention mechanism. Further, Graph Neural Networks-based methods also achieve satisfactory performance by learning multi-scale dynamics (Pfaff et al.; Fortunato et al., 2022). To improve the realism of generated results and avoid blurriness in long-term forecasts, researchers also introduce Generative Adversarial Networks (GANs) (Goodfellow et al., 2020; Esser et al., 2021; Gao et al., 2022a) and diffusion models (Rombach et al., 2022; Du et al., 2024). These advanced generative models achieve great success in prediction tasks for single, uncoupled systems. *However, all these models suffer from compounding errors in autoregressive forecasting, which degrades long-term forecast fidelity. This challenge is severely exacerbated in coupled systems, where multiple interacting models exchange and amplify their errors with each other.*

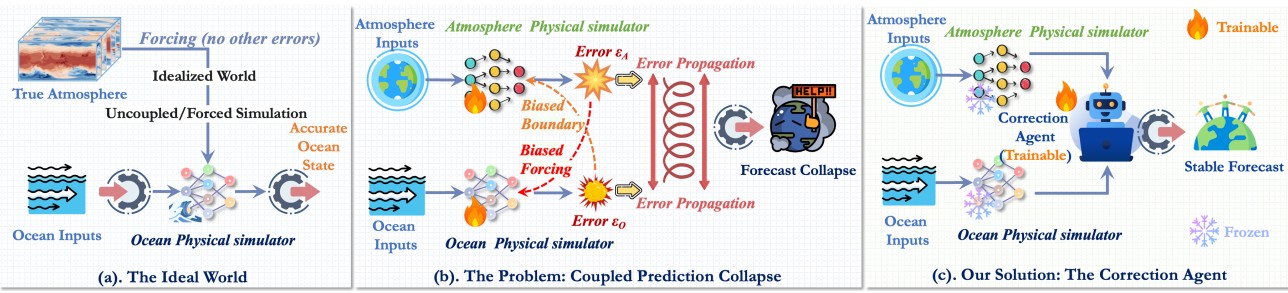

*Figure 2.* **Taming the vicious cycle of errors in coupled prediction with our `PnP-Corrector` framework. (a)** *Ideal Uncoupled Simulation*: A single simulator performs well when driven by perfect external forcing. **(b)** *Coupled Prediction Collapse*: In an autoregressive coupled mode, errors from each simulator feed into the other, leading to an exponential error growth (*Reciprocal Error Amplification*) that ultimately collapses the system. **(c)** *Our Decoupled Correction Solution*: We freeze the base simulators and train only an external Correction Agent. This agent learns to heal the systematic biases of the coupled system, steering the unstable forecast trajectory back to the ground truth to achieve long-range stability.

## 2.2. Data-Driven Earth System Forecasting

In recent years, deep learning has brought a paradigm shift to Earth system science (Reichstein et al., 2019; Jia et al., 2025). A series of milestone works, represented by Four-CastNet (Kurth et al., 2023), Pangu-Weather (Bi et al., 2023), and GraphCast (Lam et al., 2023), surpass traditional numerical models on multiple medium-range weather forecasting metrics. These models demonstrate the great potential of AI in handling high-dimensional, complex physical dynamics. Despite this, the vast majority of these SOTA models still focus on uncoupled, single-domain prediction (e.g., atmosphere-only) (Bi et al., 2023; Zhang et al., 2023; Lam et al., 2023; Gao et al., 2025b; Wu et al., 2025c; Gao et al., 2022b; Wu et al., 2023; Cui et al., 2025; Xiong et al., 2023). However, many key climate phenomena in the Earth system, such as the El Niño-Southern Oscillation (ENSO) (Ham et al., 2019; Zhu et al., 2025), are essentially driven by the complex interactions between multiple domains, primarily the ocean and atmosphere. Therefore, building coupled AI models that can stably simulate these interactions becomes the frontier of the field. Although some pioneering works explore this (Cresswell-Clay et al., 2025; Wang et al., 2024), these early attempts commonly face severe model drift and long-term instability. Their coupling mechanisms are also often model-specific and lack generality. *Thus, a general coupling framework that enables long-term, stable forecasting remains a critical challenge to be solved.*

## 2.3. Strategies for Mitigating Compounding Errors

Academia has explored various error suppression strategies, but they have fundamental limitations in coupled scenarios. Architectural approaches with physical priors (Cai et al., 2021; Wang & Yu, 2021; Wu et al., 2025a; Hess et al., 2022; Karniadakis et al., 2021; Wu et al., 2025e;b) can enhance single-model stability, but they cannot prevent the cross-contamination of errors through the coupling interface. Advanced training strategies, such as multi-step fine-tuning (Bi et al., 2023; Wu et al., 2025c; Gao et al., 2025b;a), tend to overfit the model to correcting short-term errors, causing it to collapse in true long-range forecasting as errors inevitably compound. Finally, post-processing statistical bias correction is limited by its static nature (Ross et al., 2020; Swenson & Wahr, 2006; Sanchez-Garcia et al., 2021): *it assumes stable error patterns, whereas errors in coupled systems are dynamic and state-dependent, making such static corrections ineffective and sometimes even detrimental to prediction skill.*

➠ *Our Positioning.* Existing methods fail to fundamentally address the problem of Reciprocal Error Amplification in coupled systems, creating a critical need for a general and modular solution. Our work fills this gap by proposing a new paradigm of decoupling simulation and correction, and by designing a ***plug-and-play Correction Agent***. This provides a new and extensible pathway for building stable and reliable coupled spatiotemporal forecasting systems.

## 3. Problem Formulation

Spatiotemporal forecasting aims to predict a sequence of future frames $(X_{t+1}, \ldots, X_{t+T})$ given a history of $K$ frames $(X_{t-K+1}, \ldots, X_t)$. This is equivalent to learning the conditional probability distribution $P(X_{t+1}, \ldots, X_{t+T} | X_{t-K+1}, \ldots, X_t)$. Under the Markov assumption, this problem reduces to learning a single-step predictive model, $F_\theta$, which parameterizes the conditional probability $P(X_{t+1} | X_t; \theta)$. The model is typically trained by maximizing the log-likelihood on observed sequences:

$$\max_\theta \sum_t \log P(X_{t+1} | X_t; \theta) \tag{1}$$

During inference, future frames are generated by autoregressively sampling from the learned distribution, i.e., $\hat{X}_{t+1} \sim P(\cdot | \hat{X}_t; \theta)$. Since $F_\theta$ is only an approximation of the true data distribution, errors from the sampling step accumulate, causing the condition $\hat{X}_t$ to drift from the training distribution. This phenomenon leads to ***compounding errors***.

We extend this framework to a more challenging setting: *coupled spatiotemporal forecasting*. For instance, given two interdependent systems, A and B, with states $X_t^A$ and $X_t^B$, the goal is to learn the joint conditional probability $P(X_{t+1}^A, X_{t+1}^B | X_t^A, X_t^B)$. For notational simplicity, we do not explicitly distinguish the coupling-relevant components of each subsystem from the corresponding full state variables ($X_t^A$ and $X_t^B$). This is often approximated by two interdependent models, $F^A$ and $F^B$:

$$\hat{X}_{t+1}^A \sim P(\cdot | \hat{X}_t^A, \hat{X}_t^B; \theta_A) \qquad (2)$$

$$\hat{X}_{t+1}^B \sim P(\cdot | \hat{X}_t^B, \hat{X}_t^A; \theta_B) \qquad (3)$$

In this coupled setting, error propagation is more complex. Sampling error from system A not only degrades its own future predictions but also injects a noisy condition into system B's model, distorting its predictive distribution. We term this cross-system, iterative contamination of predictive distributions *Reciprocal Error Amplification* (REA).

To mitigate REA, we introduce an independent correction network, $C_\phi$, which learns a posterior correction over the predictions of the base models. The network $C_\phi$ is trained to map the expected predictions from the base networks to a corrected, higher-fidelity output:

$$(\tilde{X}_{t+1}^A, \tilde{X}_{t+1}^B) = C_\phi \left( \mathbb{E}[\hat{X}_{t+1}^A], \mathbb{E}[\hat{X}_{t+1}^B] \right) \qquad (4)$$

where $\mathbb{E}[\cdot]$ denotes the expectation over the distributions in Eqs. (2) and (3). By minimizing the relative $L_2$-norm loss, $\mathcal{L} = ||(\tilde{X}_{t+1}^A, \tilde{X}_{t+1}^B) - (X_{t+1}^A, X_{t+1}^B)||_2$, $C_\phi$ learns to counteract the systematic biases induced by REA.

## 4. Method

To address the problem of Reciprocal Error Amplification in coupled forecasting, we propose the `PnP-Corrector`, a framework that completely decouples physical simulation from error correction. As shown in Figure 2(c), our method preserves the integrity of pre-trained physics engines by freezing their parameters and exclusively trains a correction agent to ensure long-term forecast stability. We also theoretically proved its effectiveness in terms of temporal and spatial dimensions, see the Appendix A for more details. Crucially, our framework is model-agnostic, as the physics engines can be any advanced spatiotemporal prediction models, allowing for seamless integration with other foundational models. In our implementation, the Correction Agent is an instance of our `DSLCast` architecture. To demonstrate its versatility, we also employ additional `DSLCast` instances to serve as the physics engines.

### 4.1. Decoupled Simulation-Correction Framework

The `PnP-Corrector` framework is designed to circumvent catastrophic forgetting and short-term overfitting inherent in end-to-end finetuning. Its core principle is the decoupling of responsibilities: we freeze pre-trained physics engines to simulate the core dynamics, while a correction agent is exclusively trained to counteract the systematic biases arising from their interaction. This process is divided into two stages (below is an example of two-sphere coupling).

**Stage 1: Pre-training Physics Engines** First, we independently train the atmosphere simulator $F_{\theta_A}^A$ and the ocean simulator $F_{\theta_B}^B$. In this idealized, uncoupled setting (visualized in Figure 2a), each model learns to predict the next true state $X_{t+1}$ from the current state $X_t$ and ground-truth boundary conditions $B_t$. Their respective parameters, $\theta_A$ and $\theta_B$, are optimized by minimizing their corresponding relative $L_2$-norm losses:

$$\mathcal{L}_A(\theta_A) = \mathbb{E}_{t,X} \left[ \| F_{\theta_A}^A(X_t^A, B_t^A) - X_{t+1}^A \|_2 \right] \qquad (5)$$

$$\mathcal{L}_B(\theta_B) = \mathbb{E}_{t,X} \left[ \| F_{\theta_B}^B(X_t^B, B_t^B) - X_{t+1}^B \|_2 \right] \qquad (6)$$

This stage yields two expert models that have learned the fundamental dynamics of their respective domains.

**Stage 2: Training the Correction Agent** This stage is the core of our framework, with its training paradigm illustrated in Figure 2c. We freeze the parameters $\theta = \{\theta_A, \theta_B\}$ of the physics engines and train the correction agent $C_\phi$ through an autoregressive predict-then-correct loop. At each timestep $t$, this loop consists of two steps:

*Prediction Step.* The coupled model, which we denote as a single operator $\mathcal{F}_\theta$, receives the corrected state from the previous step, $\tilde{X}_t$, and generates a preliminary but biased prediction, $\hat{X}_{t+1}$:

$$\hat{X}_{t+1} = \mathcal{F}_\theta(\tilde{X}_t) \equiv \left( F_{\theta_A}^A(\tilde{X}_t^A, \tilde{X}_t^B), \ F_{\theta_B}^B(\tilde{X}_t^B, \tilde{X}_t^A) \right) \qquad (7)$$

This step explicitly simulates the cross-contamination of errors shown in Figure 2b.

*Correction Step.* The correction agent $C_\phi$ takes this biased prediction $\hat{X}_{t+1}$ as input and outputs a corrected state $\tilde{X}_{t+1}$, effectively steering the forecast trajectory back towards the ground truth:

$$\tilde{X}_{t+1} = C_\phi(\hat{X}_{t+1}) \qquad (8)$$

The parameters $\phi$ of the correction agent are the only parameters optimized during this stage. The training objective is to minimize the relative $L_2$-norm between the corrected state and the true state:

$$\mathcal{L}_{\text{corrector}}(\phi) = \mathbb{E}_{t,X} \left\| C_\phi(\mathcal{F}_\theta(\tilde{X}_t)) - X_{t+1} \right\|_2 \qquad (9)$$

Critically, the corrected state $\tilde{X}_{t+1}$ is then fed back as the input for the next timestep, closing the loop. This forces $C_\phi$ to learn beyond correcting static, single-step biases; instead, it learns to counteract the dynamic, state-dependent error patterns of the entire coupled system.

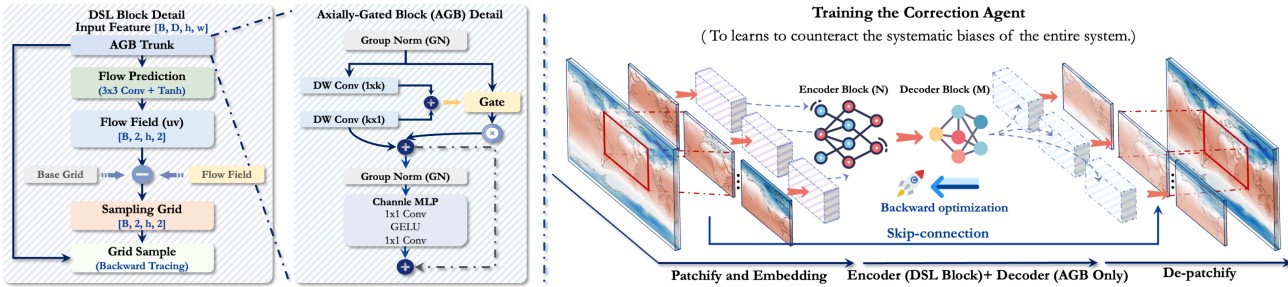

*Figure 3.* **Overview of the `DSLCast` Architecture. (Left)** Our core innovation is the Differentiable Semi-Lagrangian Advection Block (DSL-Block), which explicitly models the physical advection process. The block first predicts a Flow Field, which defines a Sampling Grid via Backward Tracing from a Base Grid. A differentiable Grid operation then warps the input features. **(Center)** The architecture is built upon the efficient Axially-Gated Block (AGB), which extracts features using parallel depthwise-separable axial convolutions ($1 \times k$ and $k \times 1$) and a learned Gate. **(Right)** The overall model is an encoder-decoder network. After the input is processed by Patchify and Embedding, the encoder (containing DSL-Blocks and AGBs) captures dynamic features, while the decoder (with AGBs only) and refinement use Skip-connection to reconstruct the high-resolution output. The model is trained end-to-end via Backward optimization.

## 4.2. The `DSLCast` Architecture

`DSLCast` is the core predictive architecture designed for the `PnP-Corrector` framework. It is a convolution-based neural network that combines an efficient, general-purpose feature extractor with a specialized block for explicitly modeling a key physical process. The design is motivated by the practical requirements of coupled Earth-system forecasting, where a model must be accurate enough to capture fine-grained spatial structures, stable enough for long autoregressive rollouts, and efficient enough to be instantiated multiple times as domain-specific engines and correction agents. Rather than relying on computationally expensive global attention, `DSLCast` emphasizes locality-aware convolutional operators, gated feature modulation, and physics-inspired feature transport. This makes it a suitable backbone for both standalone spatiotemporal prediction and the plug-and-play correction setting considered in this work.

**Overall Architecture.** As shown in Figure 3, `DSLCast` employs an encoder-decoder backbone with skip-connections. Given an input tensor $X \in \mathbb{R}^{B \times C_{in} \times H \times W}$, a convolutional embedding layer, $\text{Conv}_{patch}$, maps it into a sequence of tokens. This is then added to a fixed sinusoidal positional encoding, $P_{pos}$, to yield the initial 2D feature map $Z_0 \in \mathbb{R}^{B \times D \times h \times w}$. To further embed a geographical prior, this feature map is augmented with an Earth-Aware Latitudinal Positional Encoding, derived from a learned projection of the sine and cosine of each grid point's latitude. The encoder, $\mathcal{F}_{enc}$, is a stack of $N$ of our proposed DSL-Blocks and AGBs. The decoder, $\mathcal{F}_{dec}$, is composed of $M$ AGBs and reconstructs spatial details from the encoded feature representation. In our settings, N=16 and M=8. Finally, a transposed convolution layer $\mathcal{R}$ upsamples the decoder's output $Z_{out}$ to produce the prediction $\hat{Y} = \mathcal{R}(Z_{out})$. This prediction is then sharpened by a lightweight refinement module. The module computes a residual correction based on the concatenation of the

output $\hat{Y}$ and the original input $X$, which is applied through a zero-initialized learnable gate to ensure training stability.

**Axially-Gated Block.** The AGB is the fundamental building block of `DSLCast`, designed to efficiently capture spatial dependencies via axial decomposition. For an input feature map $F_{in}$, its computation proceeds as follows:

$$U = \text{GN}(F_{in}) \tag{10}$$
$$F_{axial} = C_{dw}^{1 \times k}(U) + C_{dw}^{k \times 1}(U) \tag{11}$$
$$G = \sigma(C_g(U)) \tag{12}$$
$$F'_{res} = F_{in} + G \odot C_{mix}(F_{axial}) \tag{13}$$
$$F_{out} = F'_{res} + \text{MLP}_C(\text{GN}(F'_{res})) \tag{14}$$

where GN is Group Normalization, $C_{dw}^{1 \times k}$ and $C_{dw}^{k \times 1}$ are parallel depthwise-separable axial convolutions, $C_g$ and $C_{mix}$ are $1 \times 1$ convolutions, $\sigma$ is sigmoid, $\odot$ denotes element-wise multiplication, and $\text{MLP}_C$ is a channel-wise MLP.

**Differentiable Semi-Lagrangian Advection Block.** The DSL-Block introduces an inductive bias for advection. For an input $F_{in}$, it first extracts features $F_{feat} = \mathcal{F}_{AGB}(F_{in})$ using an AGB trunk. A $3 \times 3$ convolution, $C_{flow}$, then operates on $F_{feat}$ to estimate a displacement flow field $\mathbf{u}$:

$$\mathbf{u} = u_{max} \cdot \tanh(C_{flow}(F_{feat})) \tag{15}$$

where $u_{max}$ is a predefined hyperparameter for the maximum displacement. We employ a semi-Lagrangian approach, performing backward tracing by subtracting the flow field from a normalized base grid $\mathbf{G}_{base}$ to compute a sampling grid:

$$\mathbf{G}_{sample} = \mathbf{G}_{base} - \mathbf{u} \tag{16}$$

Using a differentiable bilinear interpolation operator $\mathcal{W}$, we warp the feature map $F_{feat}$ according to the sampling grid, yielding $F_{warped} = \mathcal{W}(F_{feat}, \mathbf{G}_{sample})$, which reflects

the advective motion. Finally, these physically-informed features are integrated back into the network path via a separate gating mechanism:

$$G_w = \sigma(C_{gw}(F_{feat})) \tag{17}$$

$$F'_{res} = F_{in} + G_w \odot C_{mixw}(F_{warped}) \tag{18}$$

$$F_{out} = F'_{res} + \text{MLP}_C(\text{GN}(F'_{res})) \tag{19}$$

where the gate $G_w$ is conditioned on the original features $F_{feat}$ but modulates the warped features $F_{warped}$.

## 5. Experiments

In this section, we extensively evaluate the performance of `PnP-Corrector`, covering metric results, visual results, efficiency analysis, and extreme event analysis.

### 5.1. Benchmarks and Baselines

❶ **Benchmark Setup**  Our experimental benchmark is established on the ERA5 (Hersbach et al., 2020) dataset for atmospheric variables and the GLORYS12 dataset for oceanic variables. The atmospheric component comprises 69 state variables, including geopotential, specific humidity, temperature, and U/V wind components at various pressure levels, plus four key surface variables. The oceanic component includes 93 state variables, such as salinity, velocity, temperature at multiple depths, and sea surface height. In the experiments validating the extensibility of the framework, the land data across four depth levels is also sourced from ERA5. To create a standardized evaluation setting, all experiments are conducted at a 1-degree spatial resolution and a 24-hour temporal resolution. Before putting the data into the network, we normalize the variables according to their mean and std. More data processing details refer to the Appendix.

❷ **Baselines and Evaluation Protocol**  To demonstrate the universal applicability and effectiveness of our method, we select diverse baseline models to serve as the foundational physics engines. These include spatiotemporal models ***ConvLSTM*** (Shi et al., 2015) and ***SimVP*** (Tan et al., 2025); and domain-specific models for weather/ocean forecasting, namely ***GraphCast*** (Lam et al., 2023), ***Ola*** (Wang et al., 2024), and ***CirT*** (Liu et al., 2025). Our core evaluation protocol involves first training these baselines, then freezing their parameters and applying our *PnP-Corrector* as a plug-and-play enhancement. By comparing the performance of each model before and after the integration of our corrector, we can directly measure the performance gains it provides.

❸ **Evaluation Metrics**  We use Root Mean Square Error (RMSE), Mean Absolute Error (MAE), and Anomaly Correlation Coefficient (ACC) to assess forecast skill. And we use Critical Success Index (CSI) and Symmetric Extremal Dependence Index (SEDI) for extreme event evaluation.

### 5.2. Coupled Spatiotemporal Forecasting

As shown in Table 1, we report the forecasting for 162 atmosphere and ocean variables using the average results of 60 initial conditions (ICs), our `PnP-Corrector` framework consistently enhances the long-range forecast stability across all baselines. And we observe that, for many cases, this enhancement grows with the forecast lead time; for instance, the RMSE improvement for GraphCast increases from 16.45% at 120 days to 25.01% at 180 days. As depicted in Figure 4, we show RMSE and MAE results as a function of time for important variables of some representative baselines, and our `PnP-Corrector` is also competitive. Further, as shown in Table 2, we show the ACC results (average of 60 ICs) for different baselines that rank better on this metric to access the forecasting performance. We observe that the proposed `PnP-Corrector` improves the predictability of all baselines. Figure 5 visually corroborates these findings, showing that our enhanced forecasts maintain high fidelity to the ground truth over the 100-day period, which further demonstrate the effectiveness of `PnP-Corrector`.

### 5.3. Analysis of Extreme Event

Beyond average metrics, a model's ability to predict extreme events is critical for 60 ICs. We employ CSI and SEDI to evaluate its performance on extreme event. The results are shown in Figure 6, which quantitatively demonstrates the effectiveness of our `PnP-Corrector`. Specifically, when applied to the GraphCast baseline, the framework improves the CSI from 0.0743 to 0.1208 (a 62.58% increase) and boosts the SEDI score from 0.1939 to 0.2266 (a 16.86% increase). A similar trend is observed for the stronger DSLCast baseline, where the `PnP-Corrector` yields improvements of SEDI (+4.69%). This consistent enhancement across different backbone models and metrics confirms that our method significantly improves the reliability of predicting high-impact, extreme weather phenomena.

### 5.4. Analysis of Long-Range Physical Realism

Figure 7 presents a comprehensive spectral analysis of the 300-day forecast across multiple key atmospheric variables. A consistent pattern of failure emerges: without correction, all baseline models regardless of their architecture suffer from severe spectral decay at this extreme lead time, particularly at medium to high wavenumbers. This is a classic symptom of the *Reciprocal Error Amplification* we aim to solve, where compounding errors yield overly smooth and physically implausible states. In stark contrast, our model-agnostic `PnP-Corrector` universally rectifies this physical inconsistency across a diverse set of backbones (such as DSLCast, GraphCast, ConvLSTM, and CirT). The corrected spectra align remarkably well with both the ground truth and the theoretical $k^{-3}$ reference slope, even after 300 days

*Table 1.* In the global ocean and weather forecasting task, we compare the performance of our `PnP-Corrector` with 5 baselines. The average results for all 162 variables of normalized RMSE and MAE. A small RMSE (↓) and MAE (↓) indicate better performance. Relative improvements are shown with percentage. The best results are in **bold**, and the second best are with underline.

| MODELS | METRICS | | | | | | | | | |
|---|---|---|---|---|---|---|---|---|---|---|
| | 25-DAY | | 120-DAY | | 180-DAY | | 240-DAY | | 300-DAY | |
| | RMSE | MAE | RMSE | MAE | RMSE | MAE | RMSE | MAE | RMSE | MAE |
| CONVLSTM | 1.2369 | 0.9725 | 1.7426 | 1.4190 | 1.8031 | 1.4648 | 1.8229 | 1.4832 | 1.8190 | 1.4858 |
| CONVLSTM + PNP | 0.9376 | 0.7005 | 1.4525 | 1.1753 | 1.4788 | 1.1944 | 1.4632 | 1.1820 | 1.4443 | 1.1707 |
| | +24.20% | +27.96% | +16.65% | +17.17% | +17.99% | +18.46% | +19.73% | +20.31% | +20.60% | +21.20% |
| SIMVP | 1.3516 | 1.0454 | 1.3565 | 1.0947 | 1.3897 | 1.1237 | 1.3882 | 1.1260 | 1.3666 | 1.1188 |
| SIMVP + PNP | 0.7385 | 0.5160 | 1.0225 | 0.7161 | 1.0606 | 0.7540 | 1.0564 | 0.7520 | 1.0420 | 0.7495 |
| | +45.36% | +50.64% | +24.62% | +34.59% | +23.69% | +32.90% | +23.90% | +33.22% | +23.75% | +33.01% |
| GRAPHCAST | 0.7474 | 0.5173 | 1.1756 | 0.8562 | 1.4678 | 1.0884 | 1.6323 | 1.2266 | 1.6985 | 1.2844 |
| GRAPHCAST + PNP | 0.7323 | 0.5035 | 0.9822 | 0.7200 | 1.1006 | 0.8192 | 1.2261 | 0.9220 | 1.3208 | 1.0006 |
| | +2.02% | +2.67% | +16.45% | +15.91% | +25.01% | +24.73% | +24.88% | +24.84% | +22.24% | +22.10% |
| OLA | 0.9246 | 0.5684 | >100 | >100 | >100 | >100 | >100 | >100 | >100 | >100 |
| OLA + PNP | 0.7754 | 0.5506 | 33.2440 | 25.7163 | 46.0141 | 43.7024 | 46.0422 | 43.7304 | 46.0645 | 43.7536 |
| | +16.14% | +3.14% | +98.72% | +92.06% | +99.96% | +99.74% | +100.00% | +99.99% | +100.00% | +100.00% |
| CIRT | 1.6692 | 0.9218 | 2.2792 | 1.1511 | 2.3495 | 1.1909 | 2.3296 | 1.1775 | 2.2969 | 1.1532 |
| CIRT + PNP | 1.5209 | 0.8963 | 2.0074 | 1.1103 | 2.0770 | 1.1547 | 2.0504 | 1.1350 | 1.9998 | 1.0969 |
| | +8.88% | +2.76% | +11.93% | +3.54% | +11.60% | +3.04% | +11.98% | +3.61% | +12.94% | +4.88% |
| DSLCAST | 0.7286 | 0.4983 | 0.8961 | 0.6362 | 0.9328 | 0.6635 | 0.9798 | 0.7009 | 1.0012 | 0.7169 |
| DSLCAST + PNP | **0.7196** | **0.4933** | **0.8781** | **0.6292** | **0.9181** | **0.6582** | **0.9625** | **0.6946** | **0.9785** | **0.7100** |
| | +1.24% | +1.01% | +2.00% | +1.11% | +1.58% | +0.81% | +1.77% | +0.90% | +2.27% | +0.97% |

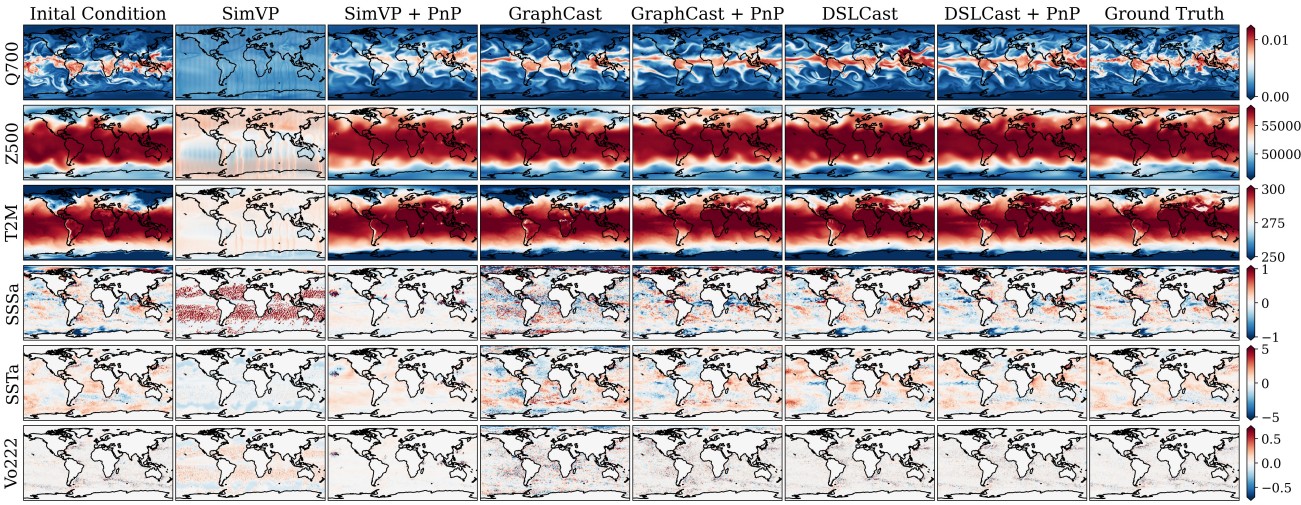

*Figure 4.* The latitude-weighted RMSE and MAE results of several important variables. Across these representative variables, `PnP-Corrector` consistently reduces errors over long lead times, highlighting its improved rollout stability.

*Figure 5.* 100-day forecast results of different models using our proposed `PnP-Corrector` framework. Our method achieves better physical consistency and yields results that are closest to the ground truth.

*Table 2.* ACC results of three of the best baselines using our `PnP-Corrector` for 93 ocean variables. A better ACC (↑) indicate better performance.

| MODELS | ACC | | | |
|---|---|---|---|---|
| | 20-DAY | 25-DAY | 35-DAY | 45-DAY |
| SIMVP | 0.2271 | 0.0778 | 0.0093 | <0 |
| SIMVP + PNP | 0.6061 | 0.5365 | 0.4175 | 0.3030 |
| | >100% | >100% | >100% | >100% |
| GRAPHCAST | 0.5177 | 0.4488 | 0.3496 | 0.2817 |
| GRAPHCAST + PNP | 0.6026 | 0.5352 | 0.4325 | 0.3600 |
| | 16.40% | 19.26% | 23.72% | 27.79% |
| OLA | 0.4980 | 0.4082 | 0.2557 | 0.1119 |
| OLA + PNP | 0.5305 | 0.4514 | 0.3353 | 0.2539 |
| | 6.51% | 10.59% | 31.11% | >100% |
| DSLCAST | 0.6387 | 0.5744 | 0.4779 | 0.4046 |
| DSLCAST + PNP | 0.6433 | 0.5789 | 0.4802 | 0.4093 |
| | 0.72% | 0.77% | 0.48% | 1.17% |

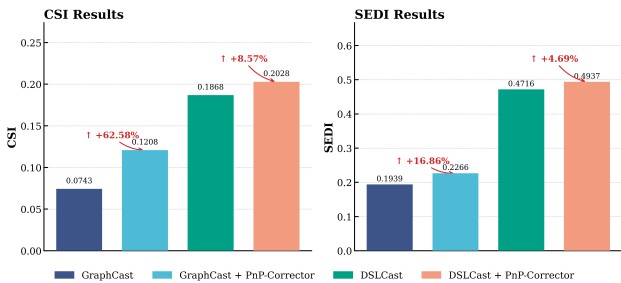

*Figure 6.* Performance evaluation on extreme event. The bar charts compare the CSI and SEDI for baselines (GraphCast and DSLCast) against their counterparts enhanced by `PnP-Corrector`.

of autoregressive rollout.

The spectral decay revealed in our analysis has direct, detrimental qualitative consequences. Figure 8 provides a clear visual example for a 100-day Mean Sea Level Pressure (MSLP) forecast using GraphCast. The standard GraphCast model exhibits significant forecast degradation, where key features like the low-pressure system over the Antarctic Peninsula become overly smooth and lose their intensity a direct visual manifestation of the spectral energy deficit. In contrast, the forecast enhanced by our `PnP-Corrector` successfully maintains the sharpness, intensity, and correct geographical position of this critical weather system, closely mirroring the ground truth.

### 5.5. Expanding Coupling Framework to More Spheres

To validate the versatility of `PnP-Corrector`, we conduct a case study on expanding this framework to more spheres. Specifically, we add the land engine, which simulates soil temperature at 4 depth levels, and conduct coupled ocean-land-atmosphere forecasts. As shown in Table 3, we report the forecasting for 166 atmosphere, ocean,

*Table 3.* Comparison on global ocean-atmosphere-land coupled forecasting task. We report the average normalized RMSE and MAE over all 166 variables for 240-day and 300-day forecasting. A small RMSE (↓) and MAE (↓) indicate better performance. Relative improvements are shown with percentage. The best results are in **bold**, and the second best are with underline.

| MODELS | 240-DAY | | 300-DAY | |
|---|---|---|---|---|
| | RMSE | MAE | RMSE | MAE |
| SIMVP | 1.3886 | 1.1282 | 1.3633 | 1.1171 |
| SIMVP + PNP | 1.1304 | 0.7979 | 1.0880 | 0.7772 |
| | +18.60% | +29.27% | +20.20% | +30.43% |
| GRAPHCAST | 1.6038 | 1.2039 | 1.6627 | 1.2559 |
| GRAPHCAST + PNP | 1.2800 | 0.9581 | 1.2606 | 0.9422 |
| | +20.19% | +20.42% | +24.18% | +24.98% |
| DSLCAST | 0.9623 | 0.6867 | 1.0003 | 0.7157 |
| DSLCAST + PNP | **0.9313** | **0.6693** | **0.9714** | **0.7005** |
| | +3.22% | +2.53% | +2.89% | +2.12% |

and land variables using the average results of 20 ICs, our `PnP-Corrector` framework consistently enhances the forecast stability. It is worth noting that we also only train the land engine and ocean-atmosphere-land correction agent for a few epochs, the coupled performance is satisfactory. Training more epochs, considering the more complex interaction of the land engine may improve the stability of the whole system. More results refer to the Appendix.

### 5.6. Experiment Setting and Efficiency Analysis

We conduct all experiments on 64 NVIDIA A100 GPUs. The training time of `DSLCast` using `PnP-Corrector` framework is about 22 hours. All baselines and our `DSLCast` are trained in the same settings using `PnP-Corrector` framework. Specifically, the ocean engine, atmosphere engine, and correction agent for ocean-atmosphere coupling are trained using 64 A100 GPUs in parallel. The ocean, atmosphere, and correction engines are trained with an initial learning rate of 0.001 under a CosineAnnealingLR schedule, in which the learning rate is gradually decreased to 0 for 200, 500, and 100 epochs, respectively. For evaluation, we use the checkpoints of the ocean engine and correction agent that achieve the lowest validation loss, together with the final checkpoint of the atmosphere engine. For the case study of expanding this framework to more spheres, the land engine and new correction agent are trained with an initial learning rate of 0.001 under a CosineAnnealingLR schedule, in which the learning rate is gradually decreased to 0 for 200 and 100 epochs, respectively. The land engine and correction for ocean-atmosphere-land coupling are trained using 32 GPUs in parallel. For evaluation, we use the checkpoints of the ocean engine and correction agent that achieve the lowest validation loss, together with the final checkpoint of the atmosphere and land engine. More details can be found in the Appendix. As shown in Table 4, our `DSLCast` has

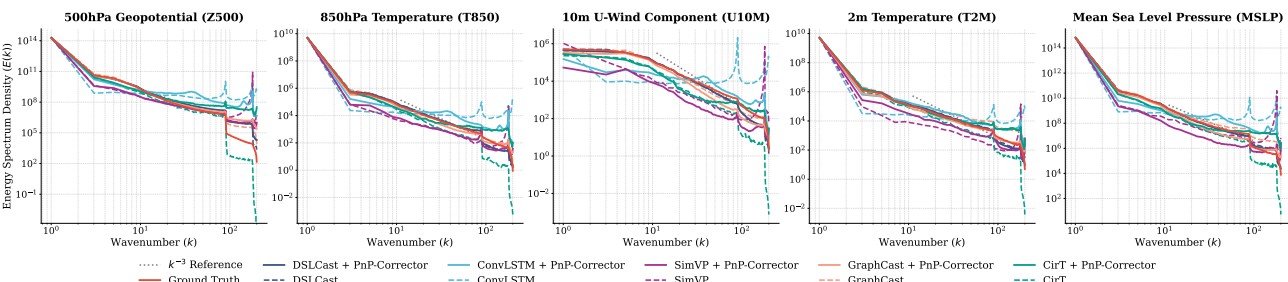

*Figure 7.* **Comprehensive Spectral Fidelity of 300-day Forecasts.** Our `PnP-Corrector` framework restores physically realistic energy spectra across multiple key atmospheric variables (Z500, T850, U10M, T2M, MSLP). The corrected models (solid lines) consistently align with the Ground Truth, demonstrating a universal improvement over the uncorrected baselines (dashed lines).

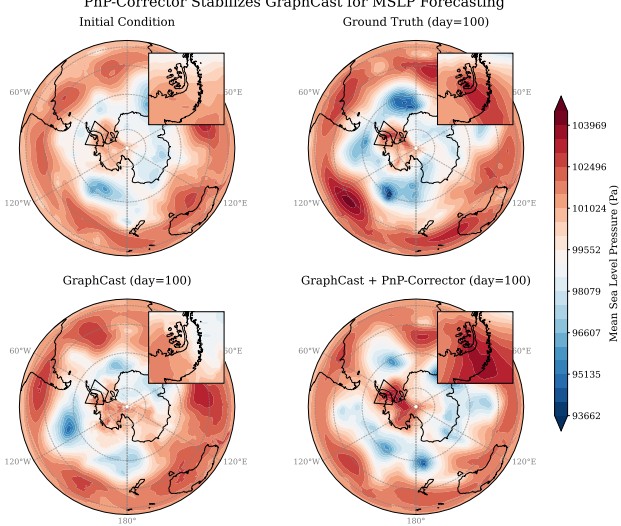

*Figure 8.* **Qualitative comparison of a** 100-**day MSLP forecast** over the Southern Hemisphere. The standard GraphCast model suffers from significant smoothing, failing to preserve the structure of the low-pressure system highlighted in the insets. Our `PnP-Corrector` counteracts this degradation, maintaining a prediction that aligns closely with the ground truth.

competitive efficiency in terms of Params and MACs.

### 5.7. Ablation Studies

To verify the effectiveness of the proposed method, as shown in Table 5, we conduct detailed ablation experiments using ERA5 and GLORYS12 dataset. We introduce the following model variants: (1) `DSLCast w/o AGB`, we remove Axially-Gated Block in the encoder of `DSLCast`. (2) `DSLCast w/o DSL`, we remove Differentiable Semi-Lagrangian Advection Block in the encoder of `DSLCast`. (3) `DSLCast`, The intact `DSLCast`. (4) `PnP-Corrector w/o Correction Agent`, we remove Correction Agent in `PnP-Corrector`. (5) `PnP-Corrector`, the intact `PnP-Corrector`. For (1), (2), and (3), we report the performance of ocean model on 93 ocean variables. For (4) and (5), we report the performance on 162 ocean and atmosphere variables. All results are based on 200-day lead time for 30 ICs of normalized RMSE and MAE. In summary, these ablation studies demon-

*Table 4.* The Params and MACs comparison.

| MODEL | PARAMS (M) | MACs (G) |
|---|---|---|
| CONVLSTM | 37.98 | 2461.09 |
| SIMVP | 38.56 | 881.85 |
| OLA | 35.22 | 142.60 |
| GRAPHCAST | 36.65 | 600.67 |
| CIRT | 61.56 | 11.07 |
| OURS | 34.76 | 572.84 |

*Table 5.* Ablation studies results, the best results are in **bold**.

| VARIANTS | RMSE | MAE |
|---|---|---|
| `DSLCast` W/O AGB | 1.0482 | 0.7724 |
| `DSLCast` W/O DSL | 1.0767 | 0.7965 |
| `DSLCast` | **0.9994** | **0.7183** |
| `DSLCast` W/O CORRECTION AGENT | 0.9444 | 0.6726 |
| `PnP-Corrector` | **0.9235** | **0.6632** |

strate that the strong performance of `PnP-Corrector` is not incidental, but arises from the synergistic interaction among its carefully designed components.

## 6. Conclusions

In this work, we identify and address the critical challenge of Reciprocal Error Amplification in coupled spatiotemporal forecasting by introducing the `PnP-Corrector`, a universal, plug-and-play correction framework. Our approach uniquely decouples the physical simulation from error correction by freezing pre-trained physics engines and exclusively training a lightweight agent to counteract systematic biases. Extensive experiments demonstrate that our framework significantly enhances the long-term stability and accuracy of diverse state-of-the-art models, proving its effectiveness and model-agnostic nature. By mitigating the vicious cycle of error amplification, the `PnP-Corrector` offers a new and robust paradigm for building reliable long-range forecasting systems for complex, interacting dynamical phenomena. While our current approach provides deterministic corrections, a key limitation is the lack of uncertainty quantification. Therefore, a goal for future work is to extend this framework to produce probabilistic forecasts, which is crucial for comprehensive risk assessment.

## Acknowledgements

This work was supported by the National Natural Science Foundation of China (42125503, 42430602).

## Impact Statement

This paper presents work whose goal is to advance the field of Machine Learning. There are many potential societal consequences of our work, none which we feel must be specifically highlighted here. In addition, we propose `PnP-Corrector`, universal correction framework for coupled spatiotemporal forecasting. In the future, we will further extent this framework to probabilistic forecasting, which is crucial for comprehensive risk assessment.

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

# A. Theoretical Analysis

To theoretically justify the stability improvements observed in Figure 1, we model the coupled forecasting process as a discrete dynamical system and analyze the error propagation bounds.

**Assumption A.1** (Lipschitz Continuity). Let $\mathcal{F}$ denote the pre-trained coupled physics simulator (representing the joint operator of $F^A$ and $F^B$) and $\mathcal{C}_\phi$ denote the correction agent. We assume $\mathcal{F}$ is $L_\mathcal{F}$-Lipschitz continuous. Given the chaotic nature of coupled systems and the phenomenon of Reciprocal Error Amplification (REA), we posit $L_\mathcal{F} > 1$. We assume $\mathcal{C}_\phi$ is $L_\mathcal{C}$-Lipschitz continuous.

**Assumption A.2** (Bounded Local Errors). Let $\epsilon_{sim} = \sup_t \|\mathcal{F}(x_t) - x_{t+1}\|$ be the single-step physical simulation error, and $\epsilon_{corr} = \sup_t \|\mathcal{C}_\phi(\mathcal{F}(x_t)) - x_{t+1}\|$ be the residual error of the correction agent trained via Eq. (9).

**Lemma A.3** (Error Bound Comparison). *For a $T$-step autoregressive rollout:*

1. *The cumulative error of the **uncorrected baseline** grows exponentially with time: $O((L_\mathcal{F})^T)$, leading to rapid forecast collapse (REA).*

2. *The cumulative error of the **PnP-Corrector** framework is governed by the effective expansion rate $\lambda = L_\mathcal{C} \cdot L_\mathcal{F}$. If the correction agent acts as a contraction mapping such that $\lambda < 1$, the error is uniformly bounded by $\frac{\epsilon_{corr}}{1-\lambda}$.*

*Proof.* Let $x_t$ be the ground truth state at time $t$.

**Case 1: Uncorrected Baseline.** The prediction follows $\hat{x}_{t+1} = \mathcal{F}(\hat{x}_t)$. Let $e_t = \|\hat{x}_t - x_t\|$ be the cumulative error. Using the triangle inequality and Lipschitz continuity:

$$
\begin{aligned}
e_{t+1} &= \|\mathcal{F}(\hat{x}_t) - x_{t+1}\| \\
&\leq \|\mathcal{F}(\hat{x}_t) - \mathcal{F}(x_t)\| + \|\mathcal{F}(x_t) - x_{t+1}\| \\
&\leq L_\mathcal{F} e_t + \epsilon_{sim}
\end{aligned}
\tag{20}
$$

Solving this recurrence with $e_0 = 0$ and $L_\mathcal{F} > 1$ yields:

$$
e_T \leq \epsilon_{sim} \sum_{k=0}^{T-1} L_\mathcal{F}^k = \epsilon_{sim} \frac{L_\mathcal{F}^T - 1}{L_\mathcal{F} - 1}
\tag{21}
$$

The term $L_\mathcal{F}^T$ confirms the exponential error explosion characteristic of REA.

**Case 2: PnP-Corrector.** The prediction follows $\tilde{x}_{t+1} = \mathcal{C}_\phi(\mathcal{F}(\tilde{x}_t))$. Let $\tilde{e}_t = \|\tilde{x}_t - x_t\|$.

$$
\begin{aligned}
\tilde{e}_{t+1} &= \|\mathcal{C}_\phi(\mathcal{F}(\tilde{x}_t)) - x_{t+1}\| \\
&\leq \|\mathcal{C}_\phi(\mathcal{F}(\tilde{x}_t)) - \mathcal{C}_\phi(\mathcal{F}(x_t))\| + \|\mathcal{C}_\phi(\mathcal{F}(x_t)) - x_{t+1}\|
\end{aligned}
\tag{22}
$$

The first term represents the propagation of previous errors through the composite function $\mathcal{C}_\phi \circ \mathcal{F}$, bounded by the product of Lipschitz constants $L_\mathcal{C} L_\mathcal{F}$. The second term is exactly the training objective of the correction agent (Eq. 9), bounded by $\epsilon_{corr}$.

$$
\tilde{e}_{t+1} \leq (L_\mathcal{C} L_\mathcal{F})\tilde{e}_t + \epsilon_{corr} = \lambda \tilde{e}_t + \epsilon_{corr}
\tag{23}
$$

By training the correction agent, we effectively optimize $\mathcal{C}_\phi$ to counteract the expansion of $\mathcal{F}$, ensuring $\lambda < 1$. Under this condition, the error converges:

$$
\lim_{T \to \infty} \tilde{e}_T \leq \frac{\epsilon_{corr}}{1 - \lambda}
\tag{24}
$$

This proves that PnP-Corrector transforms the unstable, divergent dynamics into a bounded, stable system. $\square$

# B. Dataset Details

## B.1. Dataset

In our coupled spatiotemporal forecasting experiment, the atmosphere variables are sourced from ERA5 (Hersbach et al., 2020) dataset and the ocean variables are sourced from the GLORYS12 dataset. ERA5 offers global atmosphere state, and the selected subset contains 5 variables (Z, Q, T, U, V) with 13 pressure levels (50 hPa, 100 hPa, 150 hPa, 200 hPa, 250 hPa, 300 hPa, 400 hPa, 500 hPa, 600 hPa, 700 hPa, 850 hPa, 925 hPa and 1,000 hPa) and 4 variables (U10M, V10M, T2M, MSLP) with surface level, which can be downloaded from `https://cds.climate.copernicus.eu`. GLORYS12 offers daily mean data covering latitudes between -80° and 90° from 1993 to the present, and the subset we use includes 4 depth level ocean variables (each with 23 depth levels, corresponding to 0.49 m, 2.65 m, 5.08 m, 7.93 m, 11.41 m, 15.81 m, 21.60 m, 29.44 m, 40.34 m, 55.76 m, 77.85 m, 92.32 m, 109.73 m, 130.67 m, 155.85 m, 186.13 m, 222.48 m, 266.04 m, 318.13 m, 380.21 m, 453.94 m, 541.09 m and 643.57 m), Sea salinity (S), Sea stream zonal velocity ($U_o$), Sea stream meridional velocity ($V_o$), Sea temperature ($T_o$), and 1 surface level variable Sea surface height (SSH), which can be downloaded from `https://data.marine.copernicus.eu`. For the case study in validating the extensibility of the framework, land data includes soil temperature ($T_{land}$) across 4 level is sourced from ERA5. For data partitioning, we use years from 1993 to 2020, which are 1993-2017 for training, 2018-2019 for validating, and 2020 for testing. To improve computational efficiency, we use bilinear interpolation to downsample them to 1 degree (H=181, W=360) spatial resolution. And to better adapt to the input of different architecture models, we use the data with a size of 180 × 360. The temporal resolution is 24 hours, which corresponds to 12:00 UTC for atmosphere variables, land variables, and the daily mean state for ocean variables. All the data we used are shown in Table 6 and Table 7.

*Table 6.* The data details in this work.

| Type | Full name | Abbreviation | Layers | Time | Dt | Spatial Resolution |
|---|---|---|---|---|---|---|
| Atmosphere | Geopotential | Z | 13 | 1993-2020 | 24h | 1° |
| Atmosphere | Specific humidity | Q | 13 | 1993-2020 | 24h | 1° |
| Atmosphere | Temperature | T | 13 | 1993-2020 | 24h | 1° |
| Atmosphere | Zonal wind component | U | 13 | 1993-2020 | 24h | 1° |
| Atmosphere | Meridional wind component | V | 13 | 1993-2020 | 24h | 1° |
| Atmosphere | 10 metre u wind component | U10M | 1 | 1993-2020 | 24h | 1° |
| Atmosphere | 10 metre v wind component | V10M | 1 | 1993-2020 | 24h | 1° |
| Atmosphere | 2 metre temperature | T2M | 1 | 1993-2020 | 24h | 1° |
| Atmosphere | Mean sea level pressure | MSLP | 1 | 1993-2020 | 24h | 1° |
| Ocean | Sea salinity | S | 23 | 1993-2020 | 24h | 1° |
| Ocean | Sea stream zonal velocity | $U_o$ | 23 | 1993-2020 | 24h | 1° |
| Ocean | Sea stream meridional velocity | $V_o$ | 23 | 1993-2020 | 24h | 1° |
| Ocean | Sea temperature | $T_o$ | 23 | 1993-2020 | 24h | 1° |
| Ocean | Sea surface height | SSH | 1 | 1993-2020 | 24h | 1° |

*Table 7.* The data details of case study experiment in this work.

| Type | Full name | Abbreviation | Layers | Time | Dt | Spatial Resolution |
|---|---|---|---|---|---|---|
| Land | Soil Temperature | $T_{land}$ | 4 | 1993-2020 | 24h | 1° |

## B.2. Data Preprocessing

Different atmosphere and ocean variables exhibit substantial variations in magnitude. To enable the model to concentrate on accurate simulation rather than learning the inherent magnitude discrepancies among variables, we normalize the input data prior to model ingestion. Specifically, for atmosphere variables, we calculate these statistics from an extended dataset covering the period 1993 to 2017. For ocean variables, we first compute the climatological mean of all periodic variables using data from 1993–2017 (the first 365 days of each year in the training set). The shape of the climatological mean is (365, 47, 181, 360). Specifically, 365 denotes the days, 47 denotes the number of periodic variables (ie., 23 layers Sea salinity, 23 layers Sea temperature, and Sea surface height). 181 denotes the height and 360 denotes the width. Based on this mean, Sea salinity, Sea temperature, and Sea surface height are converted into Sea salinity anomaly, Sea temperature anomaly,

and Sea surface height anomaly, respectively. We then compute the mean and standard deviation of all variables (using the anomaly fields for periodic variables) over the same 1993–2017 training period, and use these statistics for subsequent normalization. For land variables used in the case study of expanding the framework to more spheres, we also calculate these statistics from an extended dataset covering the period 1993 to 2017. Each variable thus possesses a dedicated mean and standard deviation. Before inputting data into the model, we normalize the data by subtracting the corresponding mean and dividing the respective standard deviation. For the 'nan' values of land, we fill them with zero before inputting the data into the model.

## C. Experiments Details

### C.1. Evaluation Metrics

We utilize five metrics, RMSE (Root Mean Square Error), MAE (Mean Absolute Error), ACC (Anomaly Correlation Coefficient), CSI (Critical Success Index), and SEDI (Symmetric Extremal Dependence Index) to evaluate the forecasting performance, which can be defined as:

$$\text{RMSE}(\mathcal{J}, t) = \sqrt{\frac{\sum_{i=1}^{N_{\text{lat}}} \sum_{j=1}^{N_{\text{lon}}} L(i) \left( \hat{\mathbf{A}}_{ij,t}^{\mathcal{J}} - \mathbf{A}_{ij,t}^{\mathcal{J}} \right)^2}{N_{\text{lat}} \times N_{\text{lon}}}} \tag{25}$$

$$\text{MAE}(\mathcal{J}, t) = \frac{\sum_{i=1}^{N_{\text{lat}}} \sum_{j=1}^{N_{\text{lon}}} L(i) \left| \hat{\mathbf{A}}_{ij,t}^{\mathcal{J}} - \mathbf{A}_{ij,t}^{\mathcal{J}} \right|}{N_{\text{lat}} \times N_{\text{lon}}} \tag{26}$$

$$\text{ACC}(\mathcal{J}, t) = \frac{\sum_{i=1}^{N_{\text{lat}}} \sum_{j=1}^{N_{\text{lon}}} L(i) \hat{\mathbf{A}}'^{\mathcal{J}}_{ij,t} \mathbf{A}'^{\mathcal{J}}_{ij,t}}{\sqrt{\sum_{i=1}^{N_{\text{lat}}} \sum_{j=1}^{N_{\text{lon}}} L(i) \left( \hat{\mathbf{A}}'^{\mathcal{J}}_{ij,t} \right)^2 \times \sum_{i=1}^{N_{\text{lat}}} \sum_{j=1}^{N_{\text{lon}}} L(i) \left( \mathbf{A}'^{\mathcal{J}}_{ij,t} \right)^2}} \tag{27}$$

where, latitude-dependent weights are defined as $L(i) = N_{\text{lat}} \times \frac{\cos \phi_i}{\sum_{i'=1}^{N_{\text{lat}}} \cos \phi_{i'}}$. $\phi_i$ is the latitude at index i. The anomaly of $A$, denoted as $A'$, is computed as the deviation from its climatology, which corresponds to the long-term mean of the meteorological state estimated from multiple years of training data. RMSE, MAE, and ACC are averaged across all time steps and over non-NaN spatial grid points, providing summary statistics for each variable $\mathcal{J}$ at a given lead time $\Delta t$.

$$\text{CSI}(\mathcal{J}, t) = \frac{\text{TP}}{\text{TP} + \text{FP} + \text{FN}} \tag{28}$$

$$\text{SEDI}(\mathcal{J}, t) = \frac{\log(F) - \log(H) - \log(1 - F) + \log(1 - H)}{\log(F) + \log(H) + \log(1 - F) + \log(1 - H)} \tag{29}$$

where, true positives (TP) represent the number of instances in which the ocean state is correctly simulated, while false positives (FP) and false negatives (FN) are defined analogously. $F = \frac{\text{FP}}{\text{FP}+\text{TP}}$ is the false alarm rate, and $H = \frac{\text{TP}}{\text{TP}+\text{FN}}$ represents the hit rate.

### C.2. Model Training

For ocean-atmosphere coupling, we train baseline models and our `DSLCast` using the same settings with the PnP framework. The ocean engine, atmosphere engine, and correction agent are trained with an initial learning rate of 0.001 under a CosineAnnealingLR schedule, in which the learning rate is gradually decreased to 0 for 200, 500, and 100 epochs, respectively. The atmosphere engine receives 69 atmosphere variables at time $t$, together with 1 ocean boundary condition (Sea surface temperature, SST) at time $t$ and outputs 69 atmosphere variables at time $t+1$. The ocean engine receives 93 ocean variables (for periodic variables, we first subtract its climatological mean) at time $t$, together with 8 atmosphere boundary conditions (4 surface variables at time $t$, which include U10M, V10M, T2M, MSLP, and 4 surface variables at time $t+1$), and outputs 93 ocean variables at time $t+1$. The correction agent receives 162 predicted variables (69 atmosphere variables and 93 ocean variables) at time $t+1$, and outputs 162 corrected variables at time $t+1$ as the final forecast results.

In the case study of expanding the framework to more spheres, the land engine and new correction agent are trained with an initial learning rate of 0.001 under a CosineAnnealingLR schedule, in which the learning rate is gradually decreased to 0 for

200 and 100 epochs, respectively. More training epochs of correction agent may improve the performance, however, we just intend to demonstrate the proposed framework can be expanded to more spheres. So we only train few epochs of correction agent. The land engine receives 4 land variables at time $t$, together with 3 atmosphere boundary condition (U10M, V10M, and T2M) at time $t$ and outputs 4 land variables at time $t + 1$. And the new correction agent receives 166 predicted variables (69 atmosphere variables, 93 ocean variables, and 4 land variables) at time $t + 1$, and outputs 166 corrected variables at time $t + 1$ as the final forecast results.

### C.3. Model Inference

For inference of ocean-atmosphere coupling, we use the checkpoints of the ocean engine and correction agent that achieve the lowest validation loss, together with the final checkpoint of the atmosphere engine. The atmosphere engine first receives 69 atmosphere variables at time $t$, together with 1 ocean boundary condition (Sea surface temperature, SST) at time $t$ and outputs 69 atmosphere variables at time $t + 1$. Then the 4 predicted surface variables (U10M, V10M, T2M, MSLP) at time $t + 1$, together with 4 surface variables (U10M, V10M, T2M, MSLP) at time $t$ are used as the boundary conditions of the ocean engine. The ocean engine receives 93 ocean variables at time $t$, together with 8 atmosphere boundary conditions, and outputs 93 ocean variables at time $t + 1$. Finally, the correction agent receives 162 predicted variables (69 atmosphere variables and 93 ocean variables) at time $t + 1$, and outputs 162 corrected variables at time $t + 1$ as the final forecasted results. It is worth noting that all of the atmosphere engine, ocean engine, and correction agent receive the normalized variables. So, in the long-term rollout forecasting, since the forecasted ocean state contains normalized Sea surface temperature anomalies (SSTa), we first denormalize it to unnormalized SSTa. Then, we add the corresponding climatological mean and SSTa to get unnormalized SST. Subsequently, we normalize it to normalized SST to act as the boundary condition of the atmosphere engine for the rollout forecasting. By repeating the above process, the PnP framework can generate forecasts of arbitrary length using the autoregressive approach.

In the case study of expanding the framework to more spheres, for inference, we use the checkpoints of the ocean engine and correction agent that achieve the lowest validation loss, together with the final checkpoint of the atmosphere and land engine. The atmosphere engine first receives 69 atmosphere variables at time $t$, together with 1 ocean boundary condition (Sea surface temperature, SST) at time $t$ and outputs 69 atmosphere variables at time $t + 1$. Then the 4 predicted surface variables (U10M, V10M, T2M, MSLP) at time $t + 1$, together with 4 surface variables (U10M, V10M, T2M, MSLP) at time $t$ are used as the boundary conditions of the ocean engine. The ocean engine receives 93 ocean variables at time $t$, together with 8 atmosphere boundary conditions, and outputs 93 ocean variables at time $t + 1$. The land engine first receives 4 land variables at time $t$, together with 3 atmosphere boundary condition (U10M, V10M, T2M) at time $t$ and outputs 4 land variables at time $t + 1$. Finally, the correction agent receives 166 predicted variables (69 atmosphere variables, 93 ocean variables, and 4 land variables) at time $t + 1$, and outputs 166 corrected variables at time $t + 1$ as the final forecast results. It is worth noting that all of the atmosphere engine, ocean engine, land engine, and correction agent receive the normalized variables. So, in the long-term rollout forecasting, since the forecasted ocean state contains normalized Sea surface temperature anomalies (SSTa), we first denormalize it to unnormalized SSTa. Then, we add the corresponding climatological mean and SSTa to get unnormalized SST. Subsequently, we normalize it to normalized SST to act as the boundary condition of the atmosphere engine for the rollout forecasting. By repeating the above process, the PnP framework can generate forecasts of arbitrary length using the autoregressive approach.

## D. Additional Results of the Case Study on Expanding to More Spheres

To validate the extensibility of our framework, we conduct a case study in extending the experimental scenario from ocean-atmosphere coupling to ocean-atmosphere-land coupling. As shown in Table 8, and Figure 9, our proposed `PnP-Corrector` still improves the performance of different baselines in this more challenge coupling systems.

*Table 8.* In global ocean-weather-land forecasting task, we compare the performance of our `PnP-Corrector` with 5 baselines. The average results for all 166 variables of normalized RMSE and MAE. A small RMSE (↓) and MAE (↓) indicate better performance. Relative improvements are shown with percentage. The best results are in **bold**, and the second best are with underline.

| MODELS | METRICS | | | | | | | |
|---|---|---|---|---|---|---|---|---|
| | 120-DAY | | 180-DAY | | 240-DAY | | 300-DAY | |
| | RMSE | MAE | RMSE | MAE | RMSE | MAE | RMSE | MAE |
| CONVLSTM | 1.7108 | 1.3954 | 1.7701 | 1.4400 | 1.8069 | 1.4700 | 1.7971 | 1.4662 |
| CONVLSTM + PNP | 1.0723 | 0.7924 | 1.2114 | 0.9074 | 1.2361 | 0.9369 | 1.2573 | 0.9621 |
| | +37.32% | +43.21% | +31.56% | +36.99% | +31.59% | +36.27% | +30.03% | +34.38% |
| SIMVP | 1.3368 | 1.0824 | 1.3700 | 1.1118 | 1.3886 | 1.1282 | 1.3633 | 1.1171 |
| SIMVP + PNP | 1.0136 | 0.6815 | 1.1150 | 0.7786 | 1.1304 | 0.7979 | 1.0880 | 0.7772 |
| | +24.18% | +37.04% | +18.61% | +29.97% | +18.60% | +29.27% | +20.20% | +30.43% |
| GRAPHCAST | 1.1435 | 0.8354 | 1.4383 | 1.0657 | 1.6038 | 1.2039 | 1.6627 | 1.2559 |
| GRAPHCAST + PNP | 1.0904 | 0.8194 | 1.2257 | 0.9222 | 1.2800 | 0.9581 | 1.2606 | 0.9422 |
| | +4.64% | +1.92% | +14.78% | +13.46% | +20.19% | +20.42% | +24.18% | +24.98% |
| OLA | >100 | >100 | >100 | >100 | >100 | >100 | >100 | >100 |
| OLA + PNP | 31.9352 | 27.5780 | 38.2754 | 36.0165 | 38.2840 | 36.0245 | 38.3089 | 36.0505 |
| | +98.67% | +91.14% | +99.97% | +99.78% | +100.00% | +100.00% | +100.00% | +100.00% |
| CIRT | 2.2144 | 1.1144 | 2.2921 | 1.1602 | 2.3045 | 1.1697 | 2.2562 | 1.1354 |
| CIRT + PNP | 1.8228 | 0.9998 | 1.9134 | 1.0520 | 1.9152 | 1.0528 | 1.8770 | 1.0231 |
| | +17.69% | +10.28% | +16.52% | +9.32% | +16.89% | +10.00% | +16.81% | +9.89% |
| DSLCAST | 0.8809 | 0.6271 | 0.9172 | 0.6534 | 0.9623 | 0.6867 | 1.0003 | 0.7157 |
| DSLCAST + PNP | **0.8530** | **0.6097** | **0.8927** | **0.6399** | **0.9313** | **0.6693** | **0.9714** | **0.7005** |
| | +3.17% | +2.77% | +2.67% | +2.07% | +3.22% | +2.53% | +2.89% | +2.12% |

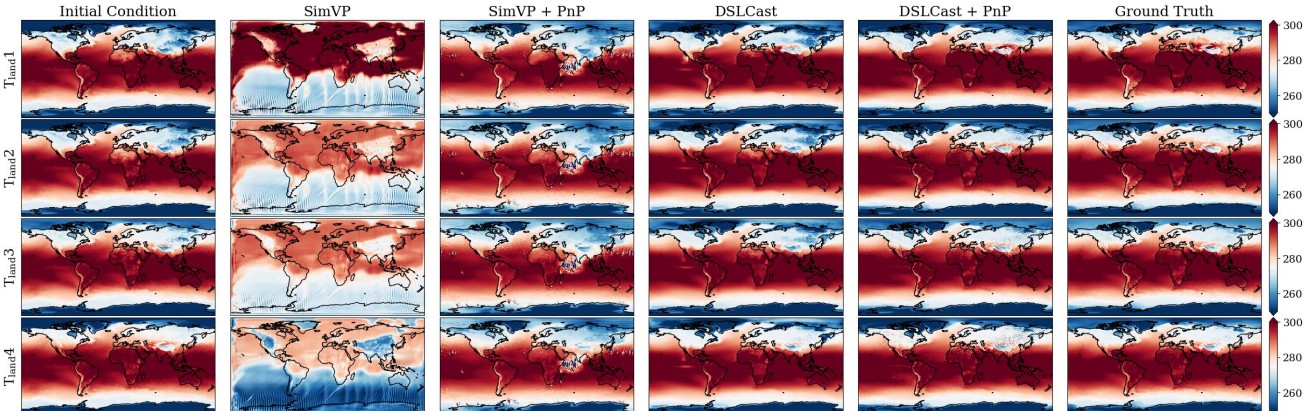

*Figure 9.* In the case study of expanding the `PnP-Corrector` framework to more spheres, we show 100-day forecast results of different models using our proposed `PnP-Corrector` framework for land variables. Our method still achieves better physical consistency and yields results that are closest to the ground truth.

# E. Additional Results

Additional quantitative results of ocean-atmosphere coupling (RMSE and MAE for important pressure level and depth level) are shown in Figure 10 and Figure 11. RMSE and MAE represent the average performance for 60 ICs, starting from Jan. 1, 2020, and the interval of each IC is 1 day. It can be seen that our proposed `PnP-Corrector` framework improves long-term forecasting stability for all baselines. Additional visual results are presented in Figure 12, Figure 13, Figure 14, and Figure 15, covering 16 important variables across forecasting time from 120 to 300 days. Collectively, these additional results further demonstrate the efficacy of the proposed `PnP-Corrector` and `DSLCast`.

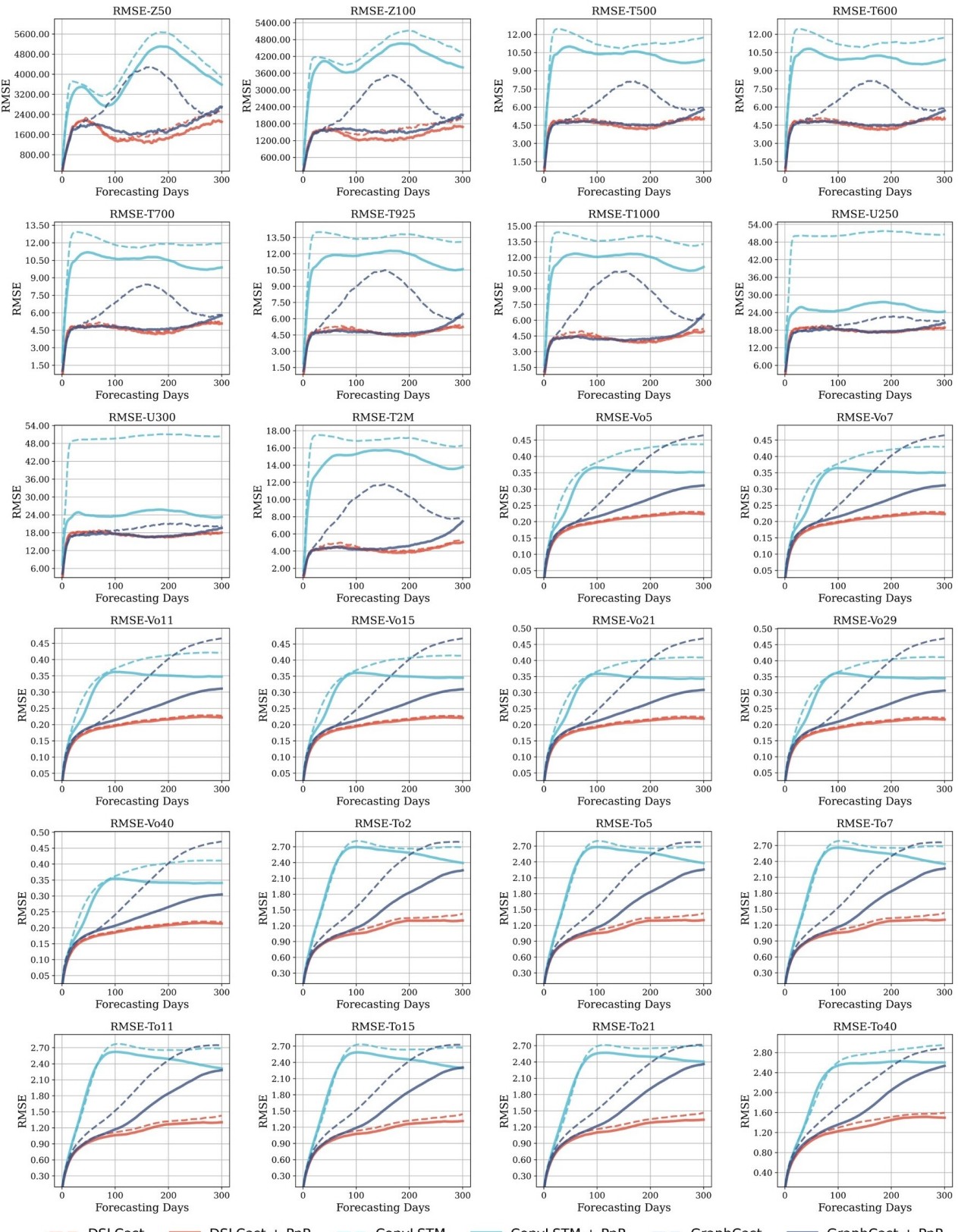

*Figure 10.* The latitude-weighted RMSE (lower is better) results of several important atmosphere and ocean variables. The corrected models that use `PnP-Corrector` framework (solid lines) achieve lower RMSE, demonstrating a universal improvement over the uncorrected baselines (dashed lines).

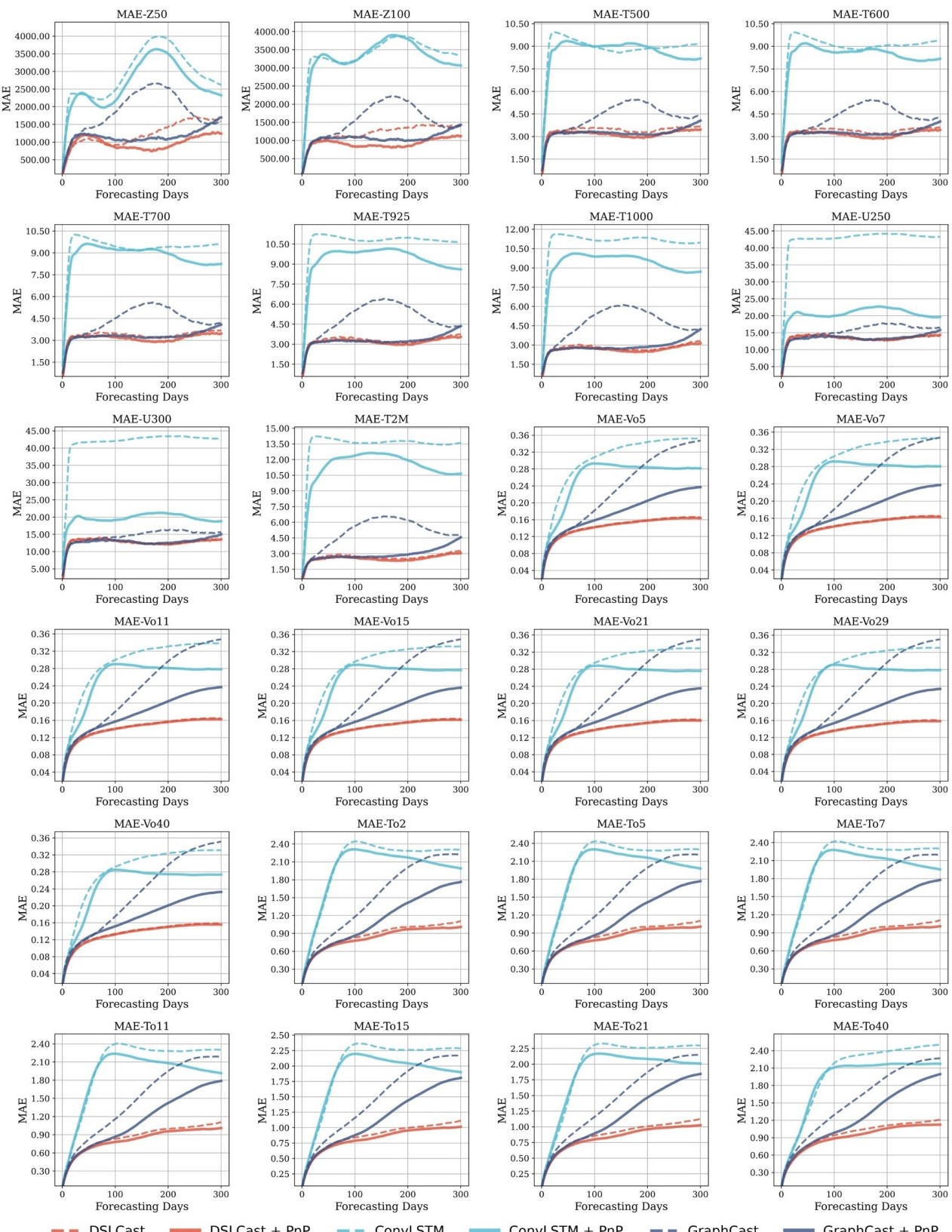

*Figure 11.* The latitude-weighted MAE (lower is better) results of several important atmosphere and ocean variables. The corrected models that use `PnP-Corrector` framework (solid lines) achieve lower MAE, demonstrating a universal improvement over the uncorrected baselines (dashed lines).

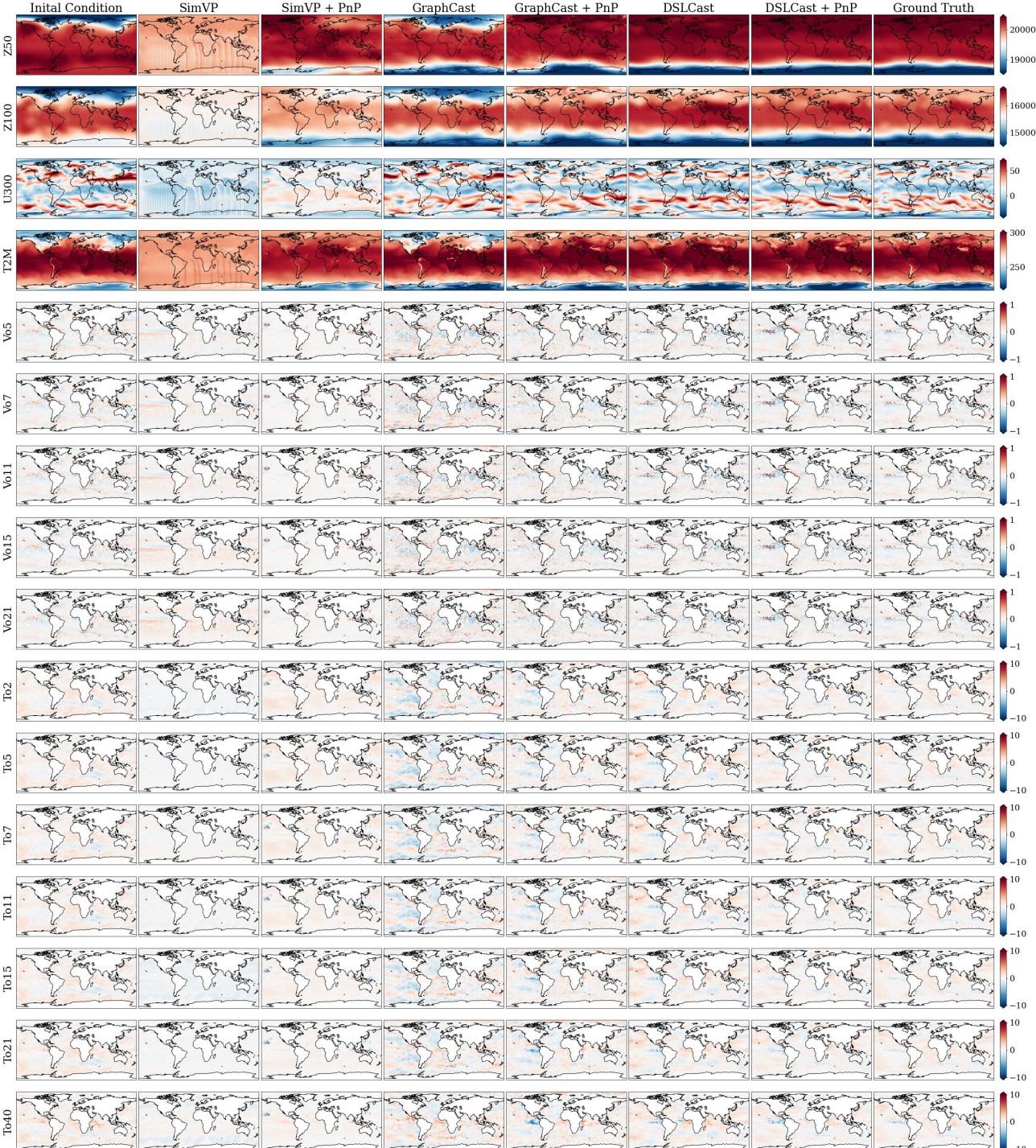

*Figure 12.* 120-day forecasting results of different models.

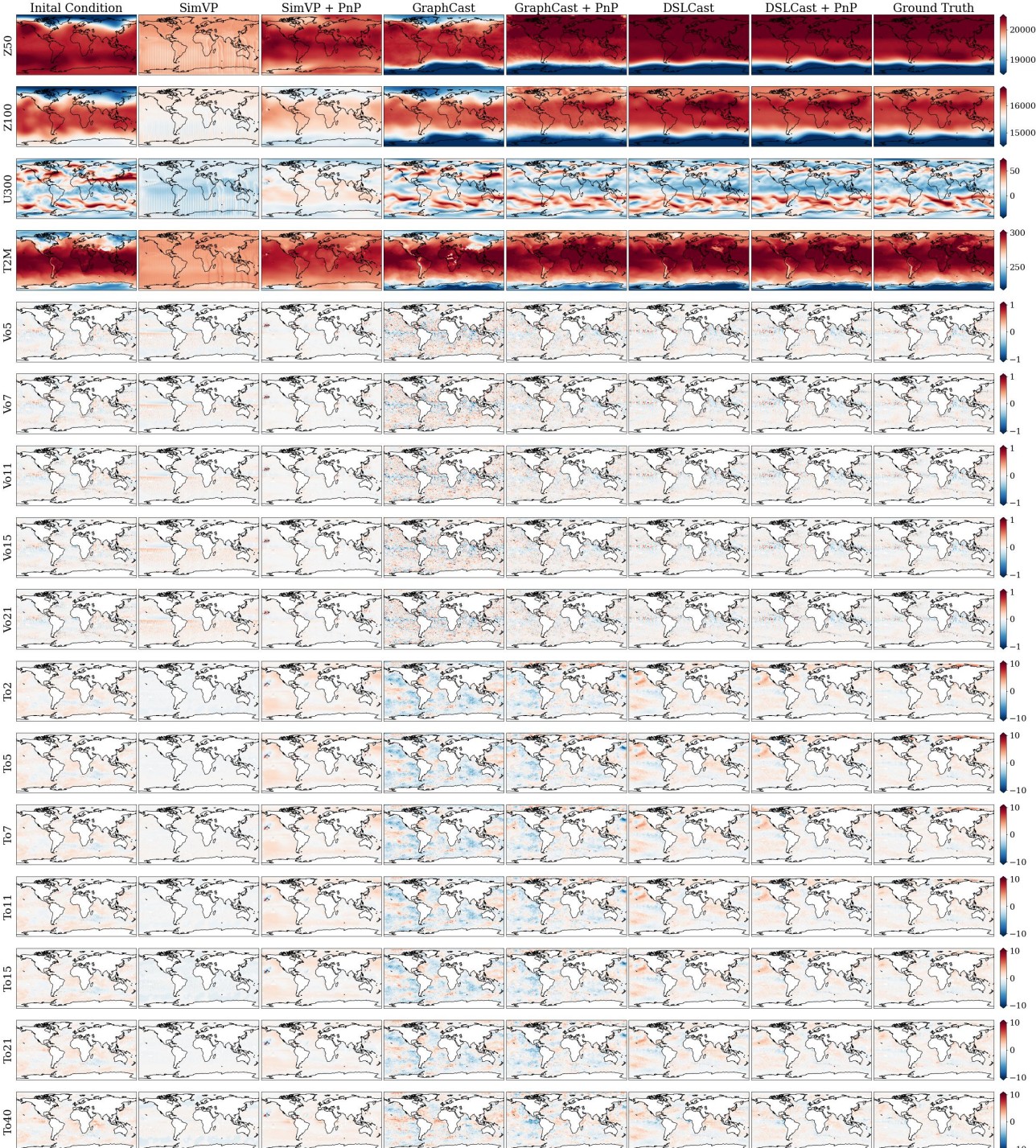

*Figure 13.* 180-day forecasting results of different models.

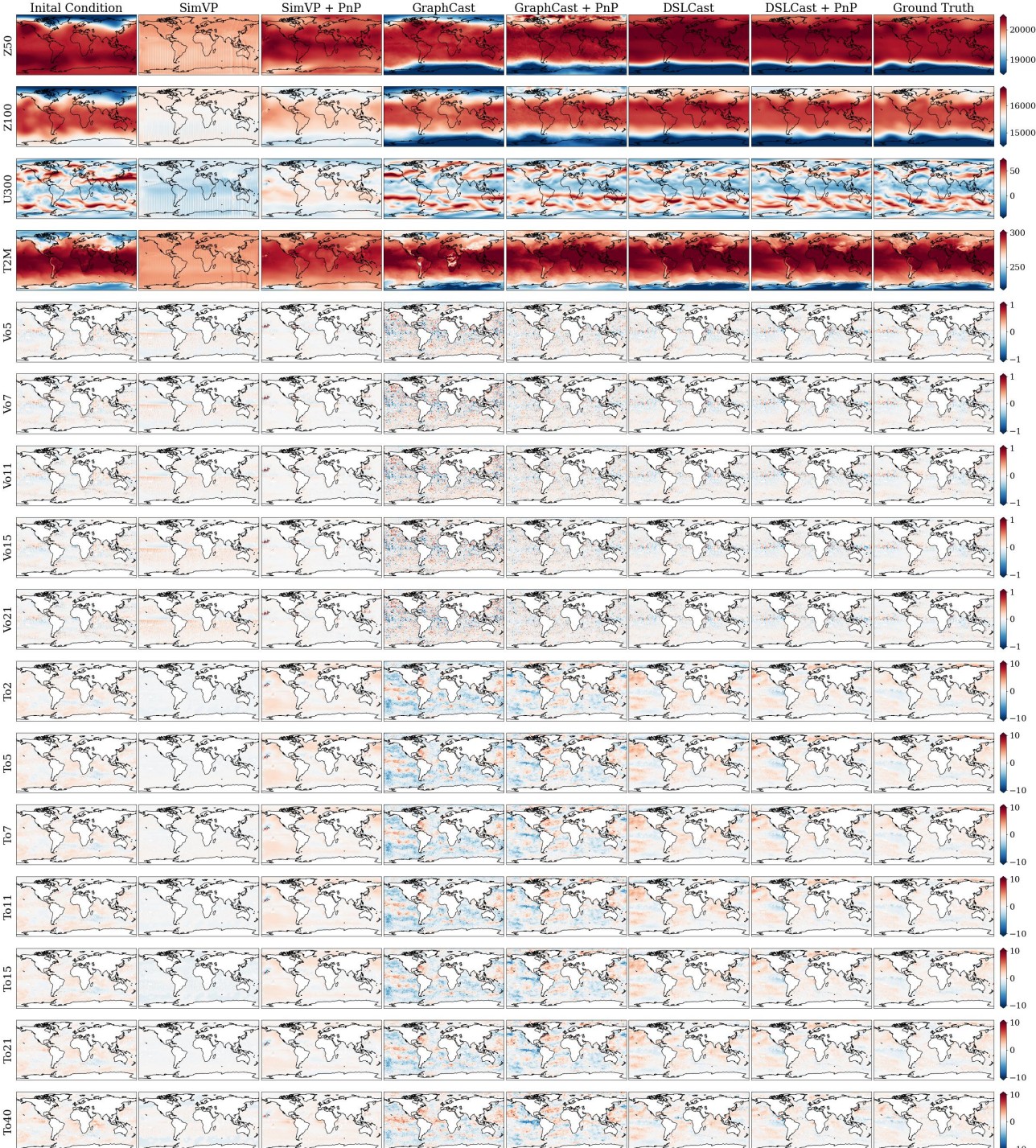

*Figure 14.* 240-day forecasting results of different models.

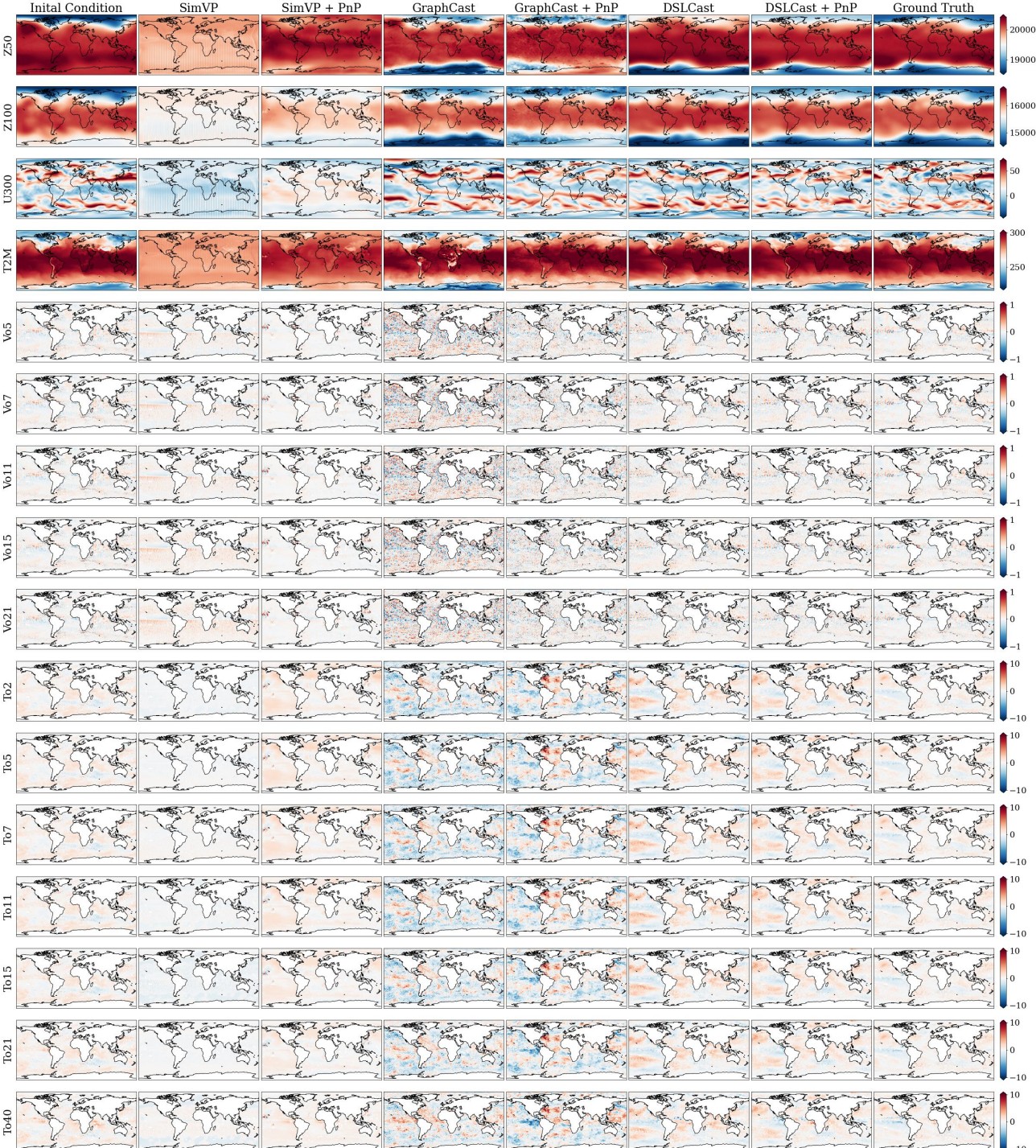

*Figure 15.* 300-day forecasting results of different models.

