# OpenReview forum: "PnP-Corrector: A Universal Correction Framework for Coupled Spatiotemporal Forecasting"
_ICML.cc/2026/Conference — ICML 2026 regular_

### Official Review · Reviewer_XmoG · 2026-03-12

**Soundness:** 3
**Presentation:** 4
**Significance:** 3
**Originality:** 3
**Overall Recommendation:** 4
**Confidence:** 3

**Summary:**

The paper presents a universal framework to correct autoregressive error build up in coupled forecasting systems as well as a semi-Lagrangian (SL) based NN that encodes the advection inductive bias within its feature maps to forecast the coupled ocean-atmosphere system.  The paper demonstrates that their corrector can be plugged into any pre-trained model to alleviate the error buildup and if used with their proposed SL-based model, the accuracy is the best. Comparisons of parameter count and FLOPs suggests that their new forecasting system is comparable to other models but with better accuracy, demonstrating it's potential

**Compliance With Llm Reviewing Policy:**

Affirmed.

**Final Justification:**

Most of my concerns were addressed. I maintain my evaluation and lean towards accepting the paper.
In terms of weaknesses: I think the era5 0.25deg evaluation could be more comprehensive (for ex, 10 ICs is very small compared to how models are generally evaluated). The 1 hour model evaluation is also understandably limited due to the very short rebuttal time. For autoregressive baselines, I'm unsure if the rebuttal answered my question. I notice 1 fig that compares simvp w finetuning and w PnP for 1 variable - I was looking for all models (Im unsure if there is a non finetuned version of GraphCast on GC, but if not, then GraphCast can be dropped) finetuned with 1-N steps rollout and, alternatively, just PnP used to compare the spectra. The rollout can also be detached through the pushforward trick to make N very large (as mentioned in the 1st review). If I missed this, I will clarify in the AC/Reviewer discussion period.

**Key Questions For Authors:**

- I couldn't tell the spectra apart - maybe it would be good to focus on 2 or 3 models and the bad models in the appendix. But it looks like the corrector is smoothing the prediction. This is also how autoregressive finetuning works in chaotic systems. It might make a good case to also finetune on some N-step rollout without the corrector and see what the difference. And also if the corrector on top of N-step rollout makes any difference. (The rollout could be detached through the pushforward trick if memory constraints).
- Figure 4 is dense again, maybe it is a good idea to show a couple of models and GT and point to differences in the long term rollout. Or maybe some spatiotemporal statistics as a function of time as well; Minor: graphcast is misspelled in this Fig.
- the SL model itself seems to be a high performing model. It would be good to compare against another advection based model (ClimODE for example)
- Could the corrector be used just for ERA5?
- For comparing models, it would be good to also report activation memory and weights (optimizer) memory (or atleast the complexity). The main issue with scaling these models to high resolution is the intermediate activation which necessitates spatial parallelism. For the SL model, it would be nice to understand how this scales with HW of the image.
- [Minor] The best models could be bolded in the table results and second best underlined so its easy to follow

**Limitations:**

Some discussion on predictability vs instability would be useful. Error compounding for chaotic systems is amplified by the fact that some processes are not predictable at the time step used and hence must be learnt either through spectral adjustments or probabilistic training.

**Strengths And Weaknesses:**

Strengths:
* The paper is written well, motivates the problem well, introduces simple but effective strategies to mitigate sources of long-term error
* The corrector module can be used with any pre-trained model
* The SL-based model stands on its own as a better model architecture

Weakensses:
* The experiments focus on coarse spatiotemporal resolution - while this is standard for ocean forecasting, for atmosphere and in general other coupled spatiotemporal systems, the resolution can be quite high. It is my understanding that the high wavenumbers are the major reason for autoregressive instability for these models (due to artifacts or just unpredictable scales that eventually diverge). To truly test the capabilities of the corrector, it might have been good to see it in action in 0.25deg 1hour ERA5.
* A key baseline may be missing - autoregressive finetuning
* Some of the visualizations are not clear (too crowded) to tease apart what is happening with the corrector or the SL model (especially the spectra)
* Advection backbones are not novel; see https://arxiv.org/pdf/2404.10024 as another possible baseline.

---

> ### Author Rebuttal · Authors · 2026-03-31
>
> Dear Reviewer XmoG,
>
> We are truly grateful for the time you have taken to review our paper and your insightful review. Here we respond to your questions point by point.
>
> > Q1. Test on 0.25deg 1hour ERA5.
>
> A1. We evaluate DSLCast on medium-range weather forecasts with 69 ERA5 variables. Due to the limited rebuttal time, we use 0.25° 6hour data. We train all baselines (params same with Table 3) on data from 1993 to 2017 for 10 epochs and report the average results over 20 initial conditions in 2019. DSLCast also achieves better performance on higher resolution data for medium-range forecasts.
>
> #RMSE (the smaller the better)
> |Model|6-h|1-day|10-day|
> |---|:-:|:-:|:-:|
> |SimVP|0.21|0.93|1.34|
> |DSLCast|0.09|0.17|0.68|
>
> > Q2. Some of the visualizations are not clear.
>
> A2. Thank you for your insightful suggestion. See our reply A4 and A5 for you.
>
> > Q3. Comparison with ClimODE.
>
> A3. We add comparison with ClimODE for ocean engine. The ClimODE is trained same as DSLCast. We report the mean results on SSTa, a key variable for marine heatwaves, over 20 initial conditions. Our DSLCast is better than ClimODE (see https://anonymous.4open.science/r/PnP_rebuttal-FFBD/ssta.jpg).
>
>
> > Q4. I couldn't tell the spectra apart - maybe it would be good to focus on 2 or 3 models and the bad models in the appendix. But it looks like the corrector is smoothing the prediction. This is also how autoregressive finetuning works in chaotic systems. It might make a good case to also finetune on some N-step rollout without the corrector and see what the difference. And also if the corrector on top of N-step rollout makes any difference.
>
> A4. We now keep only the stronger baselines in the main-text spectra comparison and focus on the evolution of T2M over time (see https://anonymous.4open.science/r/PnP_rebuttal-FFBD/spectra.jpg), and others are moved to the appendix for clarity. The results show that our corrector better restores physically realistic energy spectra and alleviates oversmoothing, leading to more stable long-term forecasts. We also compare with N-step rollout finetuning of engines without/with corrector (average results of 10 ICs for 30-day forecasts, see https://anonymous.4open.science/r/PnP_rebuttal-FFBD/finetune_spectral.jpg): N-step finetune tends to produce smoother predictions (larger spectral deviation) in chaotic systems and thus larger error accumulation over long horizons. In contrast, our original corrector restores sharper spectra and improves long-term stability.
>
> > Q5. Figure 4 is dense again. Add spatiotemporal statistics as a function of time as well; graphcast is misspelled in this Fig.
>
> A5. Thanks for your insightful suggestion. We selected some representative baselines for visual comparison, and we also show spatiotemporal statistics as a function of time for important variables. And see https://anonymous.4open.science/r/PnP_rebuttal-FFBD/visual.jpg.
>
> > Q6. Could the corrector be used just for ERA5?
>
> A6. Yes. First, for multi-sphere forecasting, since ERA5 contains data from multiple spheres (e.g., land and atmosphere), we include land data in the coupled forecasting setting to demonstrate the extensibility of our corrector (see our reply A4 for Reviewer Epqm). If using only ERA5 for land-atmosphere coupling, the task would be even simpler. Second, for single-sphere forecasting, our proposed DSLCast can also be used for ERA5 alone (see our reply A1 for you).
>
> > Q7. For comparing models, report activation memory and weights memory. For SL, test how this scales with HW of the image.
>
> A7. We report the activation memory and weights (optimizer) memory below. Due to the character limit, we compare with the best baseline GraphCast, others lay in the revision, and our DSLCast is competitive. For DSLCast, we also show the scales with HW of the image, our method is feasible to extend to high-resolution data. For example, our DSLCast can be trained on 0.25deg data (720×1440), see our reply A1 for you.
>
> |Model_H_W|weights memory(GB)|activation memory(GB)|
> |---|:-:|:-:|
> |GraphCast_180_360|0.67|8.80|
> |DSLCast_180_360|0.53|8.69|
> |DSLCast_360_720|0.55|9.38|
> |DSLCast_720_1440|0.60|12.12|
>
>
> > Q8. The best models could be bolded in the table results and second best underlined.
>
> A8. We have made the modification (see in https://anonymous.4open.science/r/PnP_rebuttal-FFBD/tab1.jpg).
>
> > Q9. Some discussion on predictability vs instability.
>
> A9. We fully agree with your point of view and will add discussion in the revision. We also believe that too large prediction step leads to some processes unpredictable, thus causing errors to accumulate and amplify, leading to the collapse of the physical field and, consequently, instability. Using spectral adjustments or probabilistic training will restore unresolved details and represent uncertainty more faithfully, relieving the errors caused by unpredictable processes, thus better for long-term stability.

---

> > ### Author Rebuttal · Reviewer_XmoG · 2026-04-04
> >
> > Thank you very much to the authors for all their responses. I maintain my positive assessment of the paper. The new presentation and comparison with climode is clear. For ERA5, could the authors post the RMSE vs time for a few variables (figures); Im unsure what the 69 vars were but some subset of u, v, z, t, q or something similar. Also, it would be good to show graphcast as well as a comparison.
> >
> > The reason I had asked for 1hour ERA5 is that the autoregressive stability gets systematically worse with smaller time steps (more accumulation). This doesn't change your computational cost (instead of t -> t+6, the target is at t+1). Is it possible to still train (it doesn't have to be SOTA) on 1 hour time steps? I realize the practical difficulties but given that the main contribution centered around "compounding errors", this would be a good baseline.
> >
> > I believe the autoregressive finetuning as a baseline is still outstanding.
> >
> > Other Qs have been answered satisfactorily.

---

> > > ### Author Response · Authors · 2026-04-08
> > >
> > > Thanks for your time to read our rebuttal and your thoughtful follow-up questions. We truly appreciate your positive assessment of our work. After receiving your new concerns, we immediately download the new data and conduct experiments, striving to provide you with a response as soon as possible. Here we respond to your follow-up questions point by point.
> > >
> > > > Q10. For ERA5, post the RMSE vs time comparison. Clarify what the 69 variables are. Add Graphcast for comparison.
> > >
> > > A10. Thank you for this helpful suggestion. Following GraphCast (publish on Science), we use 69 variables for medium-range weather forecast (0.25°, 6h), which contains 5 variables (Z, Q, T, U, V) with 13 pressure levels (50, 100, 150, 200, 250, 300, 400, 500, 600, 700, 850, 925 and 1,000 hPa) and 4 variables (U10M, V10M, T2M, MSLP) with surface level. The details are listed below:
> > >
> > > |Full name|Abbreviation|
> > > |---|:-:|
> > > |Geopotential|Z|
> > > |Specific humidity|Q|
> > > |Temperature|T|
> > > |Zonal wind component|U|
> > > |Meridional wind component|V|
> > > |Mean sea level pressure|MSLP|
> > > |2 metre temperature|T2M|
> > > |10 metre u wind component|U10M|
> > > |10 metre v wind component|V10M|
> > >
> > > And we include GraphCast as a baseline, evaluated on 10 initial conditions (ICs). The first IC is 00:00 Jan. 2019 UTC, with a 6h interval between ICs. GraphCast is trained under the same setting as the other baselines (see A1 for details). The results show that DSLCast remains competitive against GraphCast.
> > >
> > > #RMSE (the smaller the better)
> > > |Model|6-h|1-day|10-day|
> > > |---|:-:|:-:|:-:|
> > > |SimVP|0.210|0.920|1.340|
> > > |GraphCast|0.185|0.279|0.723|
> > > |DSLCast|0.102|0.171|0.692|
> > >
> > > We also post the RMSE vs time of 12 vital variables over 10 ICs, see in https://anonymous.4open.science/r/PnP_rebuttal-FFBD/rmse_vs_time_dt_6h.jpg, which further show that DSLCast outperforms the baselines across different variables and forecast horizons from 1 to 10 days.
> > >
> > > > Q11. Train on 1h 0.25° ERA5 data.
> > >
> > > A11. We fully agree with your point that the autoregressive stability gets systematically worse with smaller time steps, since the same forecast horizon requires more rollout steps and thus more accumulated error. This is also why works such as GraphCast use a 6h step rather than 1h for medium-range forecast, while some long-term works such as OLA use 24h step for stable rollouts. In our reply A1, we use 6h 0.25° ERA5 mainly for computational reasons. First, with 6h 0.25°, there are 1460 or 1464 samples per year. This tensor (1460, 69, 720, 1440) is nearly 390GB, and the training samples from 1993 to 2017 are about 9.5TB. Using 1h data increases this to 57 TB, which is difficult to train within the rebuttal period. And, we haven’t downloaded the 1h 0.25° data before. Second, we apologize that we do not fully understand your concern in Q1, so we just intend to demonstrate that our method can also be used for higher resolution spatiotemporal forecasts.
> > >
> > > To directly address your concern, we test on 1h 0.25° data. To keep the sample size consistent between the 6h and 1h settings, we select 00:00, 06:00, 12:00, and 18:00 as the ICs for both models. Due to limited rebuttal time, we just redownload 4 important surface variables (U10M, V10M, T2M, and MSLP). Both models are trained on 10 years (2008-2017) 1h 0.25° data (4 variables) for 10 epochs. The table below reports averaged results over 50 ICs in 2019. As shown in A1 and A10, DSLCast can extend to high-resolution data. Due to limited rebuttal time, as redownloading 1h data takes a huge time, we train only GraphCast on 1h data as an example to study the effect of smaller time steps on stability. RMSE vs time results (50 ICs) see in https://anonymous.4open.science/r/PnP_rebuttal-FFBD/small_step.jpg, which are consistent with your point that autoregressive stability worsens with smaller time steps.
> > >
> > > #RMSE
> > > |Model|6-h|1-day|10-day|
> > > |---|:-:|:-:|:-:|
> > > |GraphCast_1h|0.234|0.440|1.297|
> > > |GraphCast_6h|0.177|0.328|0.744|
> > > |DSLCast_6h|0.116|0.265|0.736|
> > >
> > > > Q12. I believe the autoregressive finetuning as a baseline is still outstanding.
> > >
> > > A12. Thanks for your insightful suggestion. For coupled forecasts, as shown in our reply A4, we acknowledge the role of autoregressive finetuning, and our PnP is better than autoregressive finetuning. For ERA5-6h, we add the finetuning baseline (50 ICs). Consistent with prior discussion, autoregressive finetuning improves medium-range performance, although it may produce smoother forecasts, which is also reflected by a slight decline in single-step accuracy. For RMSE vs time comparison, see in https://anonymous.4open.science/r/PnP_rebuttal-FFBD/rmse_vs_time_dt_6h_finetune.jpg.
> > >
> > > #RMSE
> > > |Model|6-h|1-day|10-day|
> > > |---|:-:|:-:|:-:|
> > > |DSLCast_6h|0.116|0.265|0.736|
> > > |DSLCast_6h_finetune|0.117|0.251|0.716|
> > >
> > > We hope these additional experiments and clarifications help address your concerns! We have done our utmost to add experiments. If possible, could you please give us stronger support by raising the score? We sincerely and humbly appreciate your consideration!

---

### Official Review · Reviewer_Epqm · 2026-03-12

**Soundness:** 2
**Presentation:** 3
**Significance:** 2
**Originality:** 3
**Overall Recommendation:** 2
**Confidence:** 4

**Summary:**

This paper proposes the PnP-Corrector, a plug-and-play correction framework designed to mitigate Reciprocal Error Amplification in coupled spatiotemporal forecasting systems. It decouples the physical simulation by freezing pre-trained independent physical models and training a separate correction agent to correct the systematic biases during autoregressive rollouts. Experimental results show the improved loss and performance in joint ocean-atmosphere coupled system forecasting.

**Compliance With Llm Reviewing Policy:**

Affirmed.

**Final Justification:**

The added results on simpler baselines and the three-domain setting are helpful, and they partially address my concerns. However, I still believe the paper needs a clearer physical justification of the correction mechanism and a more explicit discussion of its conservation-related limitations and design choices in the paper. I would like to keep my score.

**Key Questions For Authors:**

1. Could you provide a more rigorous physical justification for why the correction agent does not violate fundamental conservation laws during its mapping process?
2. Could you compare the DSLCast correction agent against a much simpler baseline, such as an MLP or a basic linear residual mapping, and plugged them in the backbones you used, to prove that this complex architectural design is strictly necessary?
3. Could you provide zoomed-in, regional visualizations tracking a specific, high-impact physical phenomenon over the forecast horizon? This would help prove that the correction agent preserves localized physical structures rather than just minimizing global loss through smoothing.
4. Could you test the performance of coupling more than two domains? For instance, introducing a third dependent simulator (such as a land/soil model) would much better demonstrate the framework's ability to manage reciprocal error amplification in highly complex, multi-component environments.

**Limitations:**

1. Weak motivation and contextualization of coupled systems. The authors fail to adequately explain the physical necessity and real-world stakes of coupled environmental systems. While they briefly mention phenomena like ENSO, the paper lacks a detailed discussion of how real-world physical, chemical, or biological interactions cause error propagation between dependent systems. Without this grounding, the problem feels fabricated for the sake of proposing an architecture, making the societal impact completely opaque. This is also reflected in the Impact Statement section, which dismisses any specific societal consequences.
2. The design of the Correction Agent is fundamentally a black-box residual mapping. By relying on a purely data-driven model to map intermediate predictions to the final target, there is no guarantee that the corrected states obey fundamental physical conservation laws (e.g., mass, energy, momentum). As a result, it is difficult to determine if the network has learned meaningful physical corrections or if it is simply overfitting to the training distribution. The authors must justify why a simpler, lightweight baseline (such as an MLP or a simple linear residual block) could not achieve a similar reduction in loss.
3. The presentation of the DSLCast architecture is difficult to follow and lacks robust justification. While it utilizes CV techiques like skip connections and Axially-Gated Blocks, the paper does not adequately explain why these specific modules are optimal for this domain. Furthermore, key notations are poorly defined in the main text. For example, the "ground-truth boundary conditions B_t"  are introduced without clearly specifying what variables they comprise or how they are obtained during real-world training and inference.
4. While the quantitative tables show loss improvement, the qualitative visual results do not convincingly demonstrate an improvement. The figures are dense and difficult to read. For example, such as the 90-day MSLP forecast in Figure 7, the GraphCast + PnP-Corrector still exhibits noticeable deviations in spatial patterns compared to the ground truth, suggesting the physical spatial pattern is not fully restored.

**Strengths And Weaknesses:**

The proposed idea of decoupling the diffrent physical simulations from the error correction is a pragmatic approach to avoiding catastrophic forgetting during end-to-end training and fine-tuning. The plug-and-play correction framework 's ability to integrate with various existing baselines (GraphCast, SimVP, etc.) demonstrates a good versatility.

While the modular-based concept is interesting, the paper suffers from weak motivation regarding the physical nature of the coupled systems, a lack of physical constraints in the correction mechanism, unclear architectural exposition, and marginal visual improvements in the experiments.

---

> ### Author Rebuttal · Authors · 2026-03-31
>
> Dear Reviewer Epqm,
>
> We are truly grateful for the time you have taken to review our paper and your insightful review. Here we respond to your questions point by point.
>
> > Q1. Why the correction agent does not violate fundamental conservation laws?
>
> A1. We don't treat the corrector as an independent physical solver, rather, it rectifies errors within the coupled system while keeping the pre-trained physics engine frozen. The Appendix demonstrates that by ensuring an effective expansion rate of $\lambda = L_{\mathcal{C}} \cdot L_{\mathcal{F}} < 1$, this method suppresses error amplification and maintains bounded trajectories; thus, it constitutes a stabilizing correction that preserves fundamental physical priors, rather than an arbitrary mapping.
>
> > Q2. Compare the DSLCast correction agent against a simpler baseline to prove that it is necessary.
>
> A2. We replace DSLCast with a simple ResNet to act as the correction agent. The RMSE below shows that this will lower the accuracy of the coupled forecasts. Since an overly simplistic model is incapable of extracting useful information, employing such a model as a correction agent may even amplify errors, resulting in performance inferior to that achieved without a correction agent. Therefore, a robust model is essential for serving as an effective correction agent.
>
> #RMSE (the smaller the better）
> |Model|120-day|180-day|240-day|300-day|
> |---|:-:|:-:|:-:|:-:|
> |DSLCast|0.901|0.937|0.985|1.001|
> |DSLCast+PnP_ResNet|2.971|4.317|4.876|5.085|
> |DSLCast+PnP_DSLCast|0.888|0.908|0.955|0.983|
>
> > Q3. Provide regional visualizations over the forecast horizon.
>
> A3. We provide zoomed-in regional results of TM2 over forecast horizons, see https://anonymous.4open.science/r/PnP_rebuttal-FFBD/loacl.jpg, which demonstrates that our correction agent restores localized physical structures rather than just minimizing global loss through smoothing.
>
> > Q4. Test the performance of coupling more than two domains.
>
> A4. We add the land engine, which simulates soil temperature at 4 depth levels, and conduct coupled ocean-land-atmosphere forecasts. Due to the limited rebuttal time, we only train land engine for 200 epochs and coupler for 20 epochs. The average results (whole variables) for 10 ICs demonstrates that PnP can be extended to more domains.
>
> #RMSE (the smaller the better)
> |Model|120-day|180-day|240-day|300-day|
> |---|:-:|:-:|:-:|:-:|
> |GraphCast|1.14|1.43|1.61|1.66|
> |GraphCast+PnP|1.04|1.17|1.26|1.31|
>
> > Q5. Weak motivation of coupled systems.
>
> A5. Thank you for the suggestion. Coupled Earth system forecasting is physically necessary because key subseasonal-to-seasonal predictability arises from cross-sphere interactions. ENSO is a canonical example, with impacts such as droughts, heatwaves, and floods. Accordingly, we study a physically meaningful global ocean–atmosphere forecasting setting rather than an artificial model concatenation, and evaluate both mean forecast skill and extreme-event reliability. We will revise the Introduction and Impact Statement to clarify these physical links and societal relevance.
>
> > Q6. There is no guarantee that the corrected states obey fundamental physical conservation law. Why a simpler baseline as correction agent could't achieves a similar reduction in loss.
>
> A6. We do not explicitly enforce conservation, but the corrected rollouts remain bounded (See our reply A1 for you and A2 for Reviewer r5Lb). And the exp results show that our correction agent preserves realistic spectra and relieves physical drift. Further, as mentioned in our reply A2 for you, if the correction agent lacks sufficient feature extraction capabilities, it can undermine the stability of the entire system.
>
> > Q7. Why the specific modules in DSLCast are optimal for this domain?
>
> A7. In DSLCast, AGB is designed to efficiently capture spatial dependencies via axial decomposition and DSL introduces an inductive bias for advection. The combination of them allows for the effective extraction of the Earth system's characteristics. We will add more details in the revision.
>
> > Q8. The 'B_t' is not clearly specified.
>
> A8. We have detailed these in Appendix C.2 (Model Training) and C.3 (Model Inference), and we will define them clearly in the main text of the revision.
>
> > Q9. The qualitative visual results do not convincingly demonstrate an improvement. The GraphCast + PnP-Corrector still exhibits deviations compared to GT.
>
> A9. We have refined the figures for better readability, see https://anonymous.4open.science/r/PnP_rebuttal-FFBD/visual.jpg and our reply A3 for you. As for Fig.7, GraphCast+PnP yields results that are closer to the ground truth than GraphCast. The ultimate forecast results depend on the synergistic interplay among the Ocean Engine, the Atmosphere Engine, and the Correction Agent. While the Correction Agent delivers performance improvements to GraphCast, if the backbone is limited, the system’s physical state cannot be fully restored.

---

### Official Review · Reviewer_r5Lb · 2026-03-13

**Soundness:** 3
**Presentation:** 3
**Significance:** 4
**Originality:** 3
**Overall Recommendation:** 5
**Confidence:** 2

**Summary:**

This paper studies long-range coupled spatiotemporal forecasting, with a focus on the error accumulation issue in interacting systems such as ocean-atmosphere prediction. The authors argue that in coupled settings, errors do not just accumulate within each model, but can also propagate back and forth across subsystems, which they call Reciprocal Error Amplification. To address this, they propose PnP-Corrector, a plug-and-play correction framework that freezes pre-trained simulators and trains only a lightweight correction agent. The paper also introduces DSLCast as a new backbone architecture. Experiments on coupled Earth system forecasting show that the method improves long-horizon stability across several backbones, and the paper includes metric, spectral, extreme-event, and ablation analyses.

**Compliance With Llm Reviewing Policy:**

Affirmed.

**Final Justification:**

I appreciate the authors' rebuttal. I also agree with Reviewer epqm’s opinion. Therefore, I am willing to raise my score while lowering my confidence.

**Key Questions For Authors:**

Please address the concerns showing in the section of weakness.

**Limitations:**

yes

**Strengths And Weaknesses:**

Strength

The overall idea of this paper is simple but practical. Instead of fine-tuning the whole coupled system, it separates simulation and correction, which makes the framework modular and easy to reuse. The method is tested on multiple backbones. The experiments are fairly comprehensive, especially the spectral analysis and extreme-event evaluation, which make the paper stronger than a standard benchmark-only submission. The writing is also generally clear, and the motivation is easy to follow.

Weakness

1. The idea of learning a correction module is reasonable, but REA can feel somewhat like a re-packaging of known error propagation behavior in coupled autoregressive systems.
2. The theoretical analysis is also quite idealized, since the main stability claim depends on a contraction-type condition that is assumed rather than verified in practice.
3. While the framework is presented as universal and plug-and-play, the paper does not fully discuss the adaptation cost, tuning effort, or training overhead required for different backbones.
4. The current method is deterministic and does not model forecast uncertainty, which is a meaningful limitation for Earth system applications.

---

> ### Author Rebuttal · Authors · 2026-03-31
>
> Dear Reviewer r5Lb,
>
> We are truly grateful for the time you have taken to review our paper and your insightful review. Here we respond to your questions point by point.
>
> > Q1. The idea of learning a correction module is reasonable, but REA can feel somewhat like a re-packaging of known error propagation behavior in coupled autoregressive systems.
>
> A1. Thank you for your insightful comment. We agree that autoregressive error propagation itself is not new. Our contribution is not to rename standard compounding error, but to isolate the coupled-specific failure mode in which errors are not only accumulated within each simulator, but are also exchanged across subsystems through the coupling interface and then fed back again in a closed loop. We term this mechanism Reciprocal Error Amplification (REA) to emphasize this bidirectional cross-system amplification, which is qualitatively more severe than standard single-model error accumulation and directly motivates a different solution design. Importantly, our main novelty is not the terminology itself, but the plug-and-play correction paradigm built on top of this formulation: instead of redesigning each backbone or fine-tuning the whole coupled system, we freeze the pre-trained simulators and train an external correction agent to counteract the systematic biases induced by coupling. The fact that this same framework consistently improves diverse backbones, especially at long-term forecasts, suggests that REA captures a practically relevant instability pattern in coupled forecasting.
>
> > Q2. The theoretical analysis is also quite idealized, since the main stability claim depends on a contraction-type condition that is assumed rather than verified in practice.
>
> A2. Thank you for this important comment. We agree that our current theoretical analysis is idealized, in that the main stability guarantee is derived under a sufficient contraction-type condition on the corrected dynamics, namely $\lambda = L_C L_F < 1$, rather than from a direct empirical estimation of $\lambda$ during practical rollouts. In the appendix, our goal is therefore not to claim that this condition is explicitly verified for every trajectory, but to clarify the mechanism by which the correction agent can stabilize an otherwise divergent coupled system: the uncorrected baseline admits exponential error growth, while the corrected rollout satisfies $\tilde e_{t+1} \le \lambda \tilde e_t + \epsilon_{\mathrm{corr}}$, so once the correction agent compresses the effective one-step error propagation, the long-horizon error becomes bounded instead of explosive. At the same time, our experiments provide strong indirect support for this analysis. First, across long rollouts up to 300 days, the forecast errors do not exhibit the exponential growth pattern expected from unstable reciprocal error amplification; instead, the proposed correction agent consistently reduces the errors of different baselines (See Tab. 1 of the main text). Second, we examine the error growth trends of several key variables over a period ranging from 1 to 300 days, and our correction agent effectively controlled these errors, see https://anonymous.4open.science/r/PnP_rebuttal-FFBD/theory.jpg. We will add more discussions and experimental results of different variables in the revision.
>
> > Q3. While the framework is presented as universal and plug-and-play, the paper does not fully discuss the adaptation cost, tuning effort, or training overhead required for different backbones.
>
> A3. Thank you for pointing out the parts we did not clearly articulate. Within this framework, the costs of training are primarily concentrated in three parts: Ocean Engine, Atmosphere Engine, and PnP. For different backbones, the difference in training costs mainly lies in the pretraining of engines using different backbones, and we report the Params and MACs in Table 3 (for Ocean Engine). It can be seen that our DSLCast performs competitively in terms of Params and MACs. At the correction stage, the Ocean and Atmosphere Engines are frozen, for all backbones, the proposed DSLCast acts as correction agent. As shown in Appendix C.2，for all backbones, we train the ocean engine, atmosphere engine, and correction agent for 200, 500, and 100 epochs, respectively. The training time of DSLCast using the PnP framework is about 22 hours on 64 A100 GPUs. For your convenience, we list the main results below.
>
> |Model|Params (M)|MACs (G)|
> |---|:-:|:-:|
> |ConvLSTM|37.98|2461.09|
> |SimVP|38.56|881.85|
> |OLA|35.22|142.60|
> |GraphCast|36.65|600.67|
> |CirT|61.56|11.07|
> |DSLCast|34.76|572.84|
>
>
> > Q4. The current method is deterministic and does not model forecast uncertainty, which is a meaningful limitation for Earth system applications.
>
> A4. Thanks for your insightful suggestion, we acknowledge this limitation. Future work will introduce the uncertainty in both the training and test stages to enhance its practical utility in Earth system forecasts.

---

> > ### Author Rebuttal · Reviewer_r5Lb · 2026-04-04
> >
> > Thank you for your answers. I'll keep my score.

---

> > > ### Author Response · Authors · 2026-04-04
> > >
> > > We appreciate your recognition of our work! Your constructive comments have reaffirmed the significance of our contributions. And we will revise the manuscript in accordance with your suggestions. Once the paper is accepted, we will release all of the codes in the camera ready phase, which includes data preprocessing, training, testing, and pre-trained weights, thereby making a modest contribution to the community.
> > >
> > > If appropriate, we would also be truly grateful for your further support by raising your score. Your score is crucial to us, and we sincerely and humbly appreciate your consideration!

---

### Decision · Program_Chairs · 2026-04-30

**Decision:**

Accept (regular)

**Comment:**

The average rating is 3.75 (5, 4, 2). After rebuttal, r5Lb raised to 5, XmoG maintained 4 noting "Most of my concerns were addressed", and Epqm maintained 2 while acknowledging "The added results on simpler baselines and the three-domain setting are helpful." The authors ran substantial new experiments during rebuttal: a three-domain (ocean-land-atmosphere) extension, ERA5 0.25deg at both 6h and 1h, ClimODE and ResNet ablations, and an autoregressive finetuning baseline. Epqm's remaining concern about conservation laws is reasonable but weaker here, since the corrector is a bias-correction layer on frozen physics engines rather than a standalone solver. The REA framing partly repackages known autoregressive error growth, and the high-resolution ERA5 evaluation is thin, but neither is a dealbreaker. The authors' confidential request to ignore Epqm is inappropriate, and in any case Epqm did engage within the discussion window, so their review counts normally.

Weighing all arguments, I lean towards acceptance. The contribution is a practical and well-executed method for correcting coupled physics-based forecasts, with substantial empirical gains over strong baselines. The paper would benefit from polishing for the camera-ready version, addressing presentation issues raised during the discussion, and including the new results in the final version. The remaining weaknesses are fixable rather than fundamental.